# Triadic embeddedness structure in family networks predicts mobile communication response to a sudden natural disaster

Jayson S. Jia [1,10 ✉], Yiwei Li [2,10], Xin Lu [3,4], Yijian Ning[5,6], Nicholas A. Christakis [7] & Jianmin Jia [8,9,10 ✉]

Kinship networks are a fundamental social unit in human societies, and like social networks in general, provide social support in times of need. Here, we investigate the impact of sudden environmental shock, the $M_s$ 7.0 2013 Ya'an earthquake, on the mobile communications patterns of local families, which we operationalize using anonymized individual-level mobile telecommunications metadata from family plan subscribers of a major carrier ($N = 35,565$ people). We demonstrate that families' communications dynamics after the earthquake depended on their triadic embeddedness structure, a structural metric we propose that reflects the number of dyads in a family triad that share social ties. We find that individuals in more embedded family structures were more likely to first call other family plan members and slower in calling non-family ties immediately after the earthquake; these tendencies were stronger at higher earthquake intensity. In the weeks after the event, individuals in more embedded family structures had more reciprocal communications and contacted more social ties in their broader social network. Overall, families that are structurally more embedded displayed higher levels of intra-family coordination and mobilization of non-family social connections.

[1] Faculty of Business and Economics, The University of Hong Kong, Hong Kong SAR, China. [2] Department of Marketing & International Business, Faculty of Business, Lingnan University, Hong Kong SAR, China. [3] College of Systems Engineering, National University of Defense Technology, Changsha, China. [4] School of Business, Central South University, Changsha, China. [5] School of Economics and Management, Southwest Jiaotong University, Chengdu, China. [6] Service Science and Innovation Key Laboratory of Sichuan Province, Chengdu, China. [7] Yale Institute for Network Science, Yale University, New Haven, CT, USA. [8] Shenzhen Finance Institute, School of Management and Economics, The Chinese University of Hong Kong, Shenzhen, China. [9] Shenzhen Research Institute of Big Data, Shenzhen, China. [10]These authors contributed equally: Jayson S. Jia, Yiwei Li, Jianmin Jia. ✉email: jjia@hku.hk; jmjia@cuhk.edu.cn

The ability of people to marshal support from their social networks in the face of setbacks is a critical survival mechanism, which contributed to the evolution of cooperation norms[1]. Although social networks are often studied under stable environmental conditions, exogenous shocks, such as natural disasters, have been the rule rather than exception throughout human history[2]. The regular occurrence of environmental disruption persists today[3–9]; in just the first two decades of the 21st century (2000–2019), 7348 recorded disasters (including 552 earthquakes) caused >1.23 million deaths and US $2.97 trillion of economic damage, and negatively affected the lives of 4 billion people[3]. More worryingly, the frequency of disasters is increasing compared to previous decades[3].

One particularly important source of support and resources in the face of such shocks is kinship networks. Kin and nonkin (volitional) networks may have markedly different structural and behavioral properties[10]. Kinship relations are less prone to decay[11], are less costly to maintain than friendships[12], and may have a greater impact on well-being than nonkin support, especially during times of particular hardship[13]. On the other hand, nonkin relations tend to be more reciprocal[14], offer greater emotional support[15], and provide greater information diversity[16]. How an individual interacts with their family vs. (nonfamily) friends after a sudden disaster not only signals their relative social priorities, but also the social network structural properties of the family.

Here, we use telecommunications data to investigate families' social network activation and interactions, following the exogenous shock of a natural disaster, the 2013 Ya'an earthquake in China ($M_s$ 7.0), both in terms of intra-family and extra-family social dynamics. We utilize the telecommunications data at this time to reveal what networks structures exhibit more active social behavior, and also how well family networks are integrated with the wider society. We show that family communications dynamics in response to the earthquake, to a significant degree, depended on the structure of the family network's embeddedness, i.e., "embeddedness structure", an alternative conceptualization of embeddedness that we introduce and test.

"Structural embeddedness", the notion that people's relationships are embedded within wider structures of social relations[16,17], is a classic theoretical construct in social science that underpins cooperation, trust, altruism, and relationship closeness[16–20]. Greater structural embeddedness can encourage dyadic trade[21], promote stable collaboration[20], facilitate favor exchanges[22], and determine social influence[23]. Structural embeddedness tends to be more stable than other properties of relationships, since it enmeshes multiple actors and is less under the control of a pair of individuals[24]. For instance, in a seminal study of 20 families in London, Bott[25] emphasized that the quality of relationships between spouses depended on their joint relationships with relatives, friends, and neighbors. The fundamental idea behind previous empirical findings is that the strength of dyadic relationships between two people also depends on others; in particular, shared mutual social ties. Previous research has typically conceptualized and operationalized structural embeddedness from a dyadic perspective using measures, such as "overlap parameter", the number of common friends shared by two people[19,20,24], or by considering contextually defined social factors, such as group affiliation or homophily[23].

Here, we develop and test the concept "structure of embeddedness", i.e., "embeddedness structure", based on the structure of social relationship sharing in triadic or higher-order motifs[26–28]. To operationalize the purely structural properties of embeddedness, we categorized each family based on their embeddedness structure graphs (Fig. 1C), where an edge represents whether two nodes shared at least one (nonfamily) friend, i.e., were embedded,

before the earthquake. No edge is drawn if a pair of nodes have no shared friends. In other words, we characterize embedded edges as a dichotomous construct (no thresholds are used). There are four possible types of embeddedness structure graphs for three-person families (Fig. 1C). We posit that embeddedness structure reflects the degree to which a family triad shares social "resources". This graph differs from standard social network graphs where edges usually correspond to direct links, the occurrence of communications, or tie strength. For example, a fully unembedded family (that shares no friends) may still have many intra-family communications. Likewise, unembedded family structures do not necessarily denote less extra-family social activity among members; for example, it is possible for individual family members to each have many friends, but for those friends to be separate from the other family members. In fact, in our data, the completely unembedded family structure (type 1) on average has higher degree centrality and more total phone calls, albeit fewer intra-family calls than the fully embedded family structure (type 4; Supplementary Table 1). This suggests that family embeddedness structure is a separate construct from relative sociability or dyadic relational strength.

Triadic structure of embeddedness differs from previous conceptualizations of structural embeddedness in two major ways. Firstly, our focal unit of analysis extends beyond one pair of embedded ties and concurrently considers how all three dyads in a triad share relationships with each other. Secondly, we focus on structural configurations (i.e., how relationships are embedded) rather than frequency of overlap (i.e., strength of dyadic embeddedness). Overall, we build on the idea that moving from a dyadic to triadic perspective of structural embeddedness focused on three-(or more) person network motifs offers fundamentally different insights for social network behavior[26–30]. Recent networks research moving beyond the study of two node links and toward higher-order interactions, and topographies have been able to study higher-order dependencies, model more complex link relationships, and even decompose otherwise unobservable indirect relationships[29]. By considering higher-order indirect relationships, we are able to measure node importance and structural influence in ways that "pairwise representation" paradigms cannot[29,30].

To obtain naturally triadic social network structures, we used anonymized individual-level mobile telecommunications data (call detailed records; CDR) from a major Chinese carrier for 35,565 subscribers based in the Ya'an prefecture of China who were in 11,855 three-person family plans (the data includes both intra- and extra-family plan communications). Mobile phone call data is one of the most commonly used forms of data in network research[5,8,9,31]. We use family plan membership to categorize communications, and explore convergences and divergences between family and friendship social network interactions. We refer to members of the same three-person family plan as core family members (i.e., likely an ego's closest family ties, or at least family ties who are close enough to subscribe to a family telecom plan together), and all other alters as "friends" (some of these "friends" could be kin, but on average, nonfamily plan members are less likely to be as closely related than family plan members). Compared to previous studies on family triads, which mainly relied on surveys[32], our data has two major advantages in measurement; firstly, family membership categorization is relatively objective (since national ID verification is required to qualify for family plan subscriptions); secondly, CDR reveals a detailed history of social interactions that is precisely quantifiable and not suspect to self-report or memory biases.

Our analysis, centered on the April 20th, 2013 Ya'an earthquake, focuses on voice call data, which dominated communications immediately after the earthquake (see "Methods" and

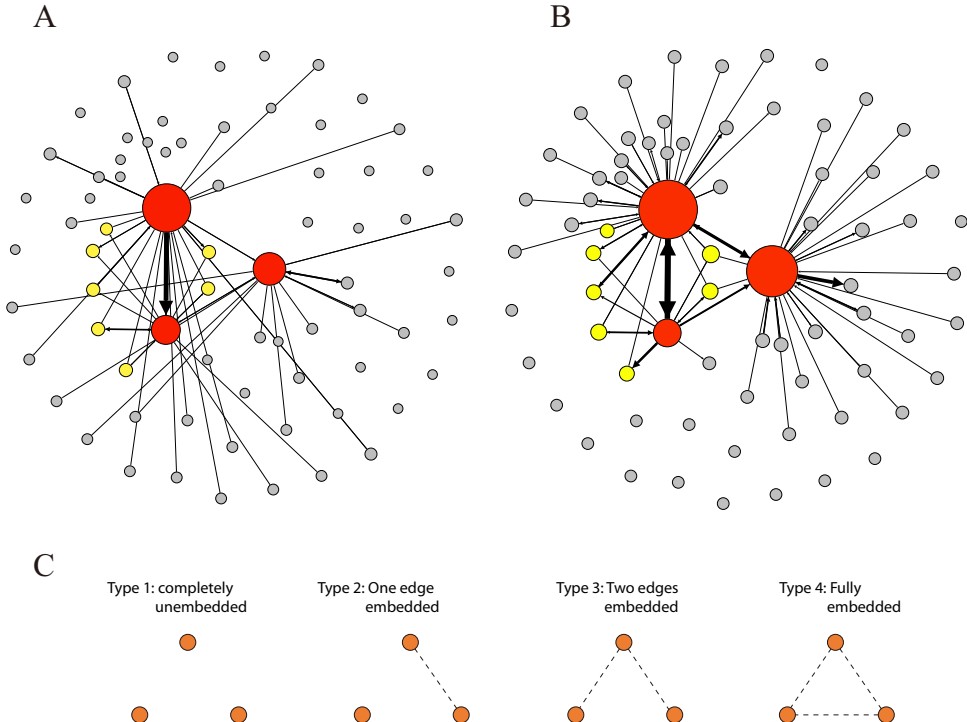

**Fig. 1 Embeddedness structure of triadic family networks. A, B** The call network of a randomly drawn family from our dataset during the week prior to the earthquake (**A**) and the first week after the earthquake (**B**) are illustrated. Node size corresponds to degree centrality; edge width to communications frequency, i.e., tie strength. Red nodes are family plan members; yellow nodes are embedded with family members; gray nodes are unembedded alters. **C** Illustrates our concept of embeddedness structure where an edge (dotted line) represents embeddedness relations (i.e., edge if two nodes share at least one friend; no edge if pair have no shared friends) rather than direct ties or communications frequency (as is the case for standard network graphs, such as **A** and **B**). There are four basic types of embeddedness structures for three-person families; a family can be fully embedded (type 4; 60.4%) or completely unembedded (type 1; 7.4%) even when the members have direct communications with each other.

Supplementary Information for discussion). Since everyone in the prefecture experienced the earthquake, the main difference in its relative impact was occurrence of physical damage or not, i.e., magnitude VIII and above or not (Supplementary Fig. 1). We used the exogenous shock of the earthquake to test whether communication behaviors in the aftermath of disaster depends on embeddedness structure and other social network constructs.

We focus on four key-dependent variables derived from behavior observed directly after the earthquake (Models 1 and 2) and also based on medium term communications (Models 3 and 4 using panel data centered on earthquake; see "Methods"). First, we examined whether or not the first call that egos made (received) after the earthquake was directed to (from) a core family member. We refer to the alter receiving an ego's first outbound call after the earthquake as the "important tie" based on the assumption that the first person someone chooses to call after experiencing a possibly traumatic disaster is emotionally and socially important to them. The assumption that choice of who to first call reflects relative importance is based on the logic of revealed preference. Second, we measured how long it took the ego to make their first outbound call to a nonfamily plan member (i.e., friend) after the earthquake; all things being equal, tempo-nonfamilyral latency of first outbound call reflects how important and urgent it is to activate one's friendship network. Third, we investigated whether embeddedness structure affected reciprocity in communications (for all relationships) after the earthquake. Reciprocity norms are critical for the maintenance of cohesion, trust, relational stability, and social capital[33], and can reflect social support and cohesiveness in both kinship and volitional groups[34]. We conservatively define communications as reciprocal if the alter of an outbound (inbound) call is also the same alter of the

next inbound (outbound) call (i.e., if the person calling the ego is also the person the ego calls next, and vice versa). Fourth, we tested the impact of embeddedness structure on centrality. Whether individuals in families with different embeddedness structures have smaller or larger active social networks after the earthquake reveals whether more embedded family structure, and the implied focus on intra-family communications, is a complement or a substitute to extra-family communications. This explores whether family embeddedness constrains or facilitates individual members' social capacity. Finally, for convergent validity, we investigate how variance of family embeddedness (an alternative operationalization reflecting imbalance in the embeddedness of the dyads in a family) affects post-earthquake social behavior. We empirically distinguish this measure from "strength of embeddedness", which is often referred to in the literature as "structural embeddedness", or simply "embeddedness", and typically measured using (dyadic) overlap parameter[21,22].

## Results

**Model-free analyses**. Throughout our analyses, we consider four basic embeddedness structures (Fig. 1C): type 1 is completely unembedded (7.4%), i.e., no family members have common friends; type 2 and type 3 are partially embedded (32.2%), i.e., some but not all dyads are embedded; and type 4 is fully embedded (60.4%), i.e., all dyads are embedded. For empirical clarity, we primarily consider the impact of the two extremes of structure (type 1 and type 4) to explore how the structure of triadic families' embeddedness affected post-earthquake communications patterns. Specifically, our empirical models contrasted completely unembedded families (i.e., type 1) against all

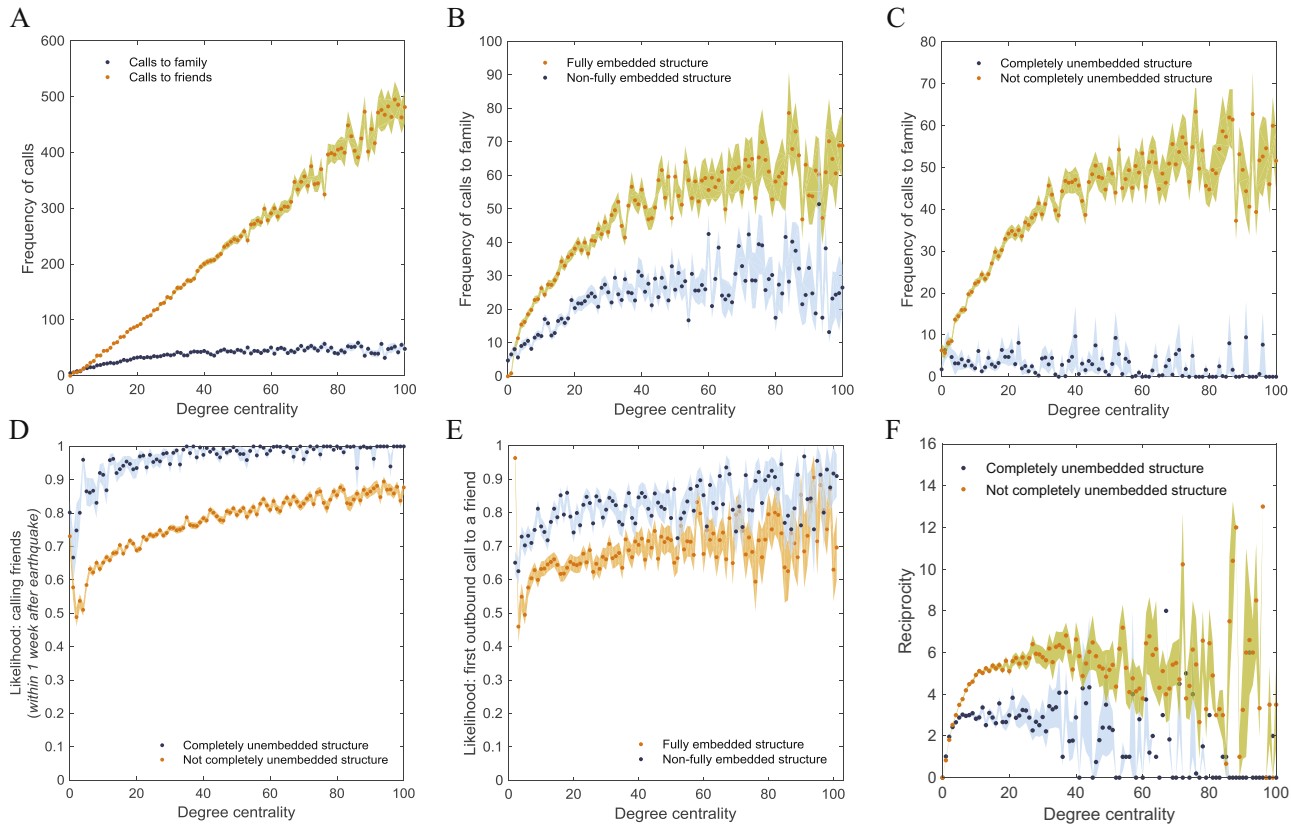

**Fig. 2 Embeddedness structure and communications frequency at different levels of degree centrality. A** Frequency of calls to friends (family) increased (was relatively flat) with degree centrality. At each level of degree centrality, **B** egos in fully embedded families made more calls to family than those in non-fully embedded families ($M = 40.68$ vs. $20.76$, $p < 0.001$). **C** Egos in completely unembedded family structures made significantly fewer calls to family than those in non-completely unembedded family structures ($M = 3.10$ vs. $35.13$, $p < 0.001$). **D** Proportion of calls to friends the first week after the earthquake is greater for egos in non-completely unembedded vs. completely unembedded family structures ($M = 0.944$ vs. $0.732$, $p < 0.001$). **E** Egos were more likely to first call a friend when they are in non-fully embedded vs. fully embedded family structures ($0.664$ vs. $0.800$, $p < 0.001$). **F** Egos are less reciprocal when they are in completely unembedded vs. non-completely unembedded family structures ($M = 1.945$ vs. $5.113$, $p < 0.001$). We observe consistent results for completely embedded vs. non-completely embedded, or completely unembedded vs. completely unembedded family structures for each dependent variable (see Supplementary Figs. 9–11 for analogous figures). $N = 35,565$ users from 11,855 three-person family plans; error bands denote 95% confidence interval.

other families, and also contrasted fully embedded families (i.e., type 4) against all other families. We made these contrasts controlling for degree centrality (Fig. 2), tie strength (Supplementary Fig. 8), and strength of family embeddedness (Supplementary Fig. 7); our results remained consistent controlling for these factors. Overall, embeddedness structure predicted both the quantity of post-disaster communications, as well as whom they were directed to. In contrast, degree centrality could only predict ego's communications frequency with friends, but not with family (Fig. 2A).

Egos in families with fully embedded structures (type 4), made more calls to family members (Fig. 2B, $M = 40.68$ vs. $20.76$, respectively, $p < 0.001$); were more likely to first call a core family member rather than a friend immediately after the earthquake (Fig. 2E, $Pr = 0.336$ vs. $0.200$, $p < 0.001$); and direct a greater proportion of post-disaster communications to family as opposed to friends ($Pr = 0.303$ vs. $0.174$, $p < 0.001$), relative to those in families that were not fully embedded (types 1–3). In other words, a fully embedded family structure corresponded to greater prioritization of intra-family social network activity.

On the other hand, egos with the conceptually opposite structure, i.e., completely unembedded family structure (type 1), made more calls to friends ($M = 209.51$ vs. $162.28$, $p < 0.001$); made fewer calls to family (Fig. 2C, $M = 3.10$ vs. $35.13$, $p < 0.001$);

were more likely to first call a friend rather than a core family member immediately after the earthquake (Fig. 2F, $0.924$ vs. $0.702$, $p < 0.001$); and direct a greater proportion of post-disaster communications to friends rather than family (Fig. 2D, $0.944$ vs. $0.732$, $p < 0.001$), relative to those in families that were not completely unembedded (types 2–4). In other words, egos who are completely unembedded in family structures had more extra-family communications, and were less likely to prioritize communications with family members.

It is notable that family embeddedness structure remained relatively stable after the earthquake (~2/3 families unchanged; Supplementary Table 8), and pre-earthquake embeddedness structure can predict post-earthquake changes in social behavior (Fig. 3, and Tables 1 and 2). In other words, family embeddedness structure is potentially an invariant that can by itself explain changes in the post-earthquake social network.

**Statistical models**. We leveraged the exogenous shock of the earthquake to test how family embeddedness structure qualified social communications response to the earthquake. Greater earthquake intensity and damage (at magnitude VIII and greater) should create greater need and demand for social coordination and support. We separated our analyses into two sets of statistical models

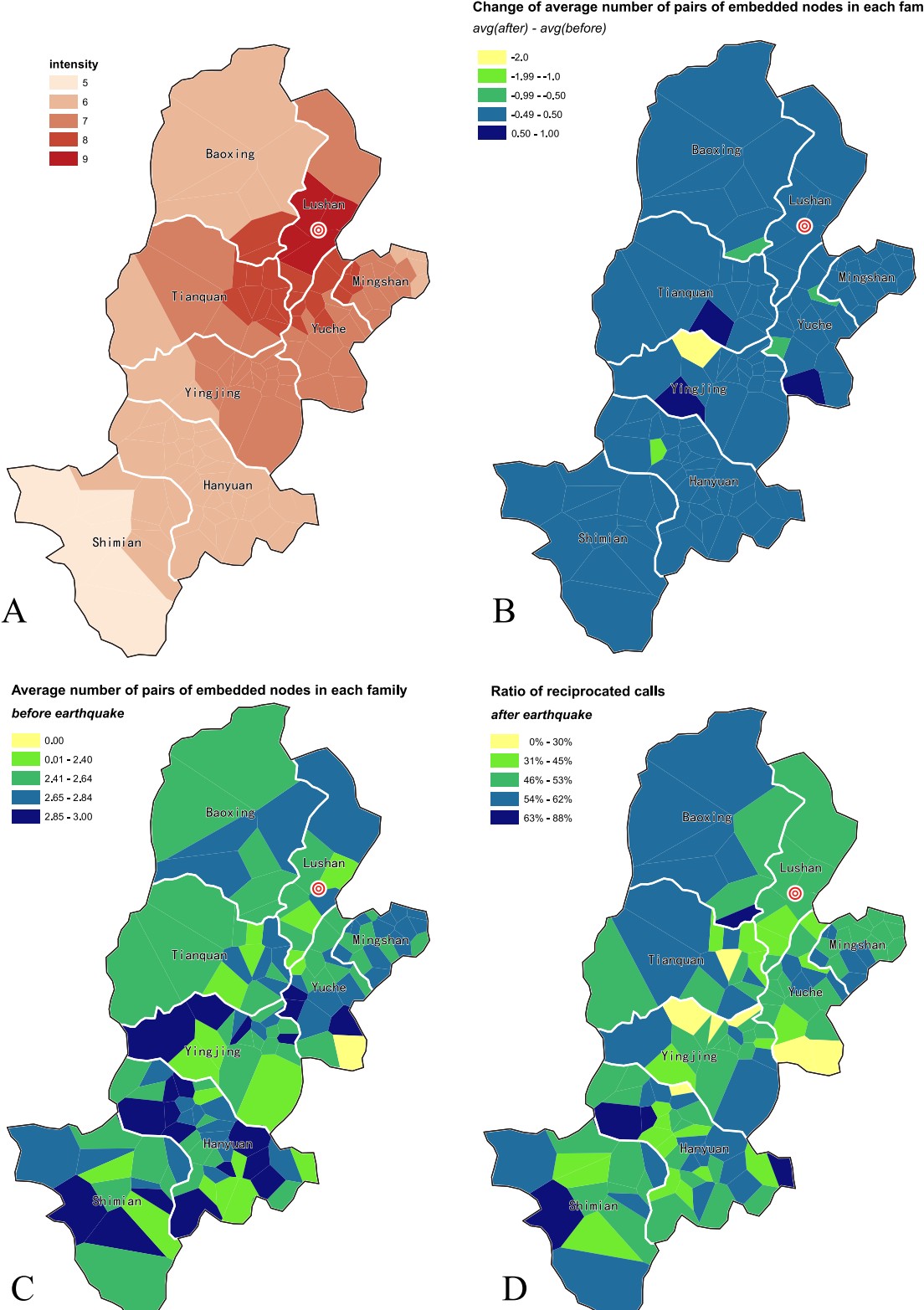

**Fig. 3 Geography of earthquake intensity and embeddedness structure. A** Earthquake intensity at each geographic locale (town/village level). **B** Change in structure of embeddedness before vs. after the earthquake at each locale: count of number of pairs of embedded nodes (i.e., embedded edges or links) in families after the earthquake is stable and similar to before the earthquake. **C** Embeddedness structure 1–4 weeks before the earthquake at each locale (average number of family dyads that share a nonfamily friend). **D** Average ratio of reciprocated calls across locales 2–5 weeks after the earthquake. Intensity is not significantly correlated with embeddedness, Pearson's $r = -0.128$, $p = 0.138$, or reciprocity, $r = -0.106$, $p = 0.222$; embeddedness and reciprocity are significantly correlated, $r = 0.505$, $p < 0.0001$. Overall, embeddedness structure is stable to the earthquake's shock and is correlated with post-earthquake reciprocity behavior.

**Table 1 Target of initial outbound calls.**

| Dependent variable = p(first outbound call is to nonfamily plan member) | Coef. | SE | z | p > \|z\| |
|---|---|---|---|---|
| (1) Ego and family all unembedded | 0.572 | 0.067 | 8.50 | <0.001*** |
| (2) Ego and family all embedded | −0.267 | 0.027 | −9.97 | <0.001*** |
| Earthquake intensity group (1 = severe) | 0.075 | 0.034 | 2.18 | 0.029* |
| (1) X Earthquake intensity group | 0.227 | 0.104 | 2.19 | 0.029* |
| (2) X Earthquake intensity group | −0.003 | 0.041 | −0.06 | 0.950 |
| ln(Degree centrality of ego) | 0.166 | 0.023 | 7.14 | <0.001*** |
| ln(Total call frequency of ego) | −0.074 | 0.018 | −4.22 | <0.001*** |
| ln(Total text frequency of ego) | 0.009 | 0.007 | 1.40 | 0.162 |
| ln(Internet usage frequency of ego) | −0.008 | 0.006 | −1.35 | 0.176 |
| ln(Phone retail price in Yuan) | −0.013 | 0.013 | −1.03 | 0.304 |
| ln(Total WeChat frequency) | <0.001 | 0.007 | 0.06 | 0.952 |
| ln(Total frequency of other instant messaging) | 0.002 | 0.006 | 0.34 | 0.734 |
| Roaming dummy (1 = traveling outside of prefecture) | −0.343 | 0.026 | −13.00 | <0.001*** |
| Rural dummy (1 = rural) | −0.087 | 0.021 | −4.20 | <0.001*** |
| Damage dummy (1 = cell towers damaged in neighborhood) | 0.267 | 0.068 | 3.90 | <0.001*** |
| Prior intra-family outbound call share | −1.847 | 0.072 | −25.65 | <0.001*** |
| (Intercept) | 0.920 | 0.091 | 10.14 | <0.001*** |

Notes: number of obs. = 26,394; model controls for random effects of families; the p values are obtained from a two-sided z-test.
Model 1 tests whether higher earthquake intensity affects the relationship between family embeddedness structure and likelihood that an ego's first outbound call after the earthquake was to a nonfamily plan member (i.e., friend). We find that, if a family is completely unembedded, its members are more likely to call a friend first after the earthquake at higher earthquake intensities. However, fully embedded family structures only had a main effect; egos in fully embedded (unembedded) families were more (less) likely to first call a family member regardless of earthquake intensity.
***p < 0.001, **p < 0.01, *p < 0.05.

**Table 2 Latency until first outbound call to a nonfamily plan member.**

| Dependent variable = hours until first outbound call to nonfamily plan member | Coef. | SD | Lower 2.5% CI | Upper 97.5% CI | $N_{eff}$ | Rhat |
|---|---|---|---|---|---|---|
| (1) Ego and family all unembedded | −26.562 | 4.834 | −36.139 | −17.217 | 15987 | 1.000 |
| (2) Ego and family all embedded | 55.705 | 4.758 | 46.359 | 64.924 | 16055 | 1.000 |
| Earthquake intensity group (1 = severe) | −16.582 | 4.723 | −25.860 | −7.295 | 16590 | 1.000 |
| (1) X Earthquake intensity group | −14.358 | 5.015 | −24.258 | −4.506 | 16895 | 1.000 |
| (2) X Earthquake intensity group | 13.890 | 4.666 | 4.679 | 22.994 | 16106 | 1.000 |
| ln(Degree centrality of ego) | −56.805 | 4.351 | −65.345 | −48.366 | 13819 | 1.000 |
| ln(Total call frequency of ego) | −34.806 | 3.720 | −41.939 | −27.701 | 12139 | 1.000 |
| ln(Total text frequency of ego) | −20.910 | 3.213 | −27.242 | −14.715 | 15452 | 1.000 |
| ln(Internet usage frequency of ego) | 2.933 | 2.976 | −2.916 | 8.682 | 12377 | 1.000 |
| ln(Phone retail price in Yuan) | −4.618 | 2.901 | −10.125 | 0.999 | 11380 | 1.000 |
| ln(Total WeChat frequency) | −3.423 | 3.501 | −10.289 | 3.325 | 14675 | 1.000 |
| ln(Total frequency of other instant messaging) | −11.889 | 3.138 | −18.166 | −5.767 | 12061 | 1.000 |
| Roaming dummy (1 = traveling outside of prefecture) | 19.607 | 4.818 | 10.413 | 28.968 | 17453 | 1.000 |
| Rural dummy (1 = rural) | 14.648 | 4.702 | 5.469 | 23.678 | 16445 | 1.000 |
| Damage dummy (1 = cell towers damaged in neighborhood) | −1.800 | 4.928 | −11.441 | 7.914 | 15578 | 1.000 |
| Prior intra-family outbound call share | 24.572 | 4.892 | 14.878 | 34.197 | 15516 | 1.000 |
| (Intercept) | 1.141 | 4.952 | −8.533 | 10.740 | 16486 | 1.000 |

Notes: number of obs. = 18,958; model controls for random effects of families; Bayesian estimations produce credible intervals (CI), which are the intervals within which the parameter values fall at some particular probability.
Model 2 tests if higher earthquake intensity affects the relationship between family embeddedness structure and how long it took egos to contact a nonfamily plan member (i.e., "friend"). A negative interaction term with (1) showed that at higher earthquake intensity, completely unembedded family structures were significantly faster in calling friends; a positive interaction term with (2) shows that at higher earthquake intensity levels, fully embedded family structures were significantly slower in calling friends. In other words, at higher earthquake intensity, people in completely unembedded families exhibited greater relative urgency in reaching out to friends, while people in fully embedded families were relatively less urgent.

(Tables 1–4 and see "Methods"): (1) models using cross sectional data of social network activation and calling behavior immediately after the earthquake (Tables 1 and 2, which test the relative impact of earthquake intensity), and (2) models using panel data of social network structural change before and after the earthquake (Tables 3 and 4, which test before vs. after effects). Finally, we use variance in the embeddedness of family relationships (variance in the overlap parameters of edges, i.e., pairs of nodes) as an alternative independent variable to explore the role of relative structural imbalance (Supplementary Tables 15–18).

**Model 1: friend or family**. We first tested if embeddedness structure predicted ego's choice of important tie, in particular, the likelihood that an ego's first outbound call after the earthquake was to a friend, rather than family. We used a logistic random effects regression model with individual families as random effects (Table 1). The interaction term between the embeddedness structures and the earthquake intensity dummy variable (magnitude VIII or greater) tested if family embeddedness structure qualified the impact of the earthquake. Control variables included dyadic embeddedness (overlap parameter) with important tie,

**Table 3 Communications reciprocity after earthquake.**

| Dependent variable = reciprocity | Coef. | Cluster SE | t | p > |t| |
|---|---|---|---|---|
| Ego and family all embedded | NA | NA | NA | NA |
| Earthquake dummy (1 = post-quake) | 0.225 | 0.034 | 6.58 | <0.001*** |
| Ego and family all embedded X earthquake dummy | 0.186 | 0.082 | 2.27 | 0.023* |
| ln(Degree centrality of ego) | −0.731 | 0.052 | −14.06 | <0.001*** |
| ln(Total call frequency of ego) | 1.293 | 0.035 | 36.97 | <0.001*** |
| ln(Total text frequency of ego) | 0.133 | 0.025 | 5.32 | <0.001*** |
| ln(Internet usage frequency of ego) | 0.015 | 0.006 | 2.69 | 0.007** |
| ln(Phone retail price in Yuan) | 0.168 | 0.035 | 4.74 | <0.001*** |
| ln(Total WeChat frequency) | −0.005 | 0.018 | −0.26 | 0.795 |
| ln(Total frequency of other instant messaging) | −0.015 | 0.014 | −1.03 | 0.303 |
| Roaming dummy (1 = traveling outside of prefecture) | −0.054 | 0.068 | −0.79 | 0.428 |
| Rural dummy (1 = rural) | 0.092 | 0.167 | 0.55 | 0.583 |
| Damage dummy (1 = cell towers damaged in neighborhood) | −1.096 | 0.338 | −3.25 | 0.001** |
| Prior intra-family outbound call share share | −0.076 | 0.170 | −0.44 | 0.657 |
| Family fixed effects | Yes | | | |

Notes: number of obs. = 49,851; time periods = 5; the p values are obtained from a two-sided t test; model controls for fixed effects of families; SE's clustered at family levels; first variable is perfectly collinear with family fixed effects, and thus was excluded from model estimation.
Model 3 tested if fully embedded families had more reciprocal communications after the earthquake. We counted total number of reciprocal communications across the span of our panel data, where a call is reciprocal if the alter of an outbound (inbound) call was also the same alter of the next inbound (outbound) call (no distinction made between family and friends). The significant positive interaction term shows that the earthquake increased quantity of reciprocal communications for people with fully embedded family structures, relative to those with completely unembedded family structures.
***p < 0.001, **p < 0.01, *p < 0.05.

**Table 4 Degree centrality after earthquake.**

| Dependent variable = ego's centrality | Coef. | Cluster SE | t | p > |t| |
|---|---|---|---|---|
| Ego and family all embedded | NA | NA | NA | NA |
| Earthquake dummy (1 = post-quake) | 0.045 | 0.022 | 2.03 | 0.043* |
| Ego and family all embedded X earthquake dummy | 0.299 | 0.052 | 5.73 | <0.001*** |
| ln(Total call frequency of ego) | 1.565 | 0.036 | 43.76 | <0.001*** |
| ln(Total text frequency of ego) | 0.407 | 0.040 | 10.22 | <0.001*** |
| ln(Internet usage frequency of ego) | −0.008 | 0.007 | −1.12 | 0.263 |
| ln(Phone retail price in Yuan) | 0.345 | 0.050 | 6.93 | <0.001*** |
| ln(Total WeChat frequency) | 0.100 | 0.032 | 3.11 | 0.002*** |
| ln(Total frequency of other instant messaging) | −0.038 | 0.020 | −1.87 | 0.061 |
| Roaming dummy (1 = traveling outside of prefecture) | −0.645 | 0.084 | −7.67 | <0.001*** |
| Rural dummy (1 = rural) | −0.014 | 0.184 | −0.07 | 0.941 |
| Damage dummy (1 = cell towers damaged in neighborhood) | −0.309 | 0.665 | −0.47 | 0.642 |
| Prior intra-family outbound call share share | −4.346 | 0.211 | −20.61 | <0.001*** |
| Family fixed effects | Yes | | | |

Notes: number of obs. = 49,851; time periods = 5; the p values are obtained from a two-sided t test; model controls for fixed effects of families; SE's clustered at family levels; first variable is perfectly collinear with family fixed effects, and thus was excluded from model estimation.
Model 4 tested if families with fully embedded structures called more people after the earthquake (degree centrality): the significant positive interaction between earthquake intensity and fully embedded family structure shows that at higher earthquake intensity, fully embedded families had greater degree centrality.
***p < 0.001, **p < 0.01, *p < 0.05.

dyadic tie strength, ego's degree centrality, ego's pre-earthquake call, text, and internet usage frequency, phone price, whether ego used instant messaging, whether ego was out of town during earthquake (roaming), whether ego lived in a rural district, and whether there was damage to cell towers in the neighborhood during earthquake. We find that, if a family is completely unembedded, higher earthquake intensity resulted in a higher chance of the ego calling a friend rather than family. However, fully embedded family structures yielded only a main effect; egos in fully embedded families were more likely to first call a family member regardless of earthquake intensity. The results are robust if the dependent variable is whether the first inbound call was from a friend or a core family member (Supplementary Table 5). In other words, embeddedness structure can also predict who first reached out to the ego after the earthquake.

A competing interpretation is that the dependent variable simply reflected whether or not the family plan members lived together. For example, if earthquake victims' first outbound call was always directed to their next closest family member who was not physically present, then our results documented the relationship between embeddedness and likelihood of family cohabitation. As a robustness check, we reran Model 1 selecting only for customers who were roaming (i.e., physically not in Ya'an, N = 3515, Supplementary Table 7), and obtained the same pattern of results. There was still a significant positive interaction between earthquake intensity and completely unembedded family structure. This finding is consistent with previous research showing that greater physical proximity and interactions, for example from shared residence, is associated with more, not less, mobile phone communications[7,12,35,36].

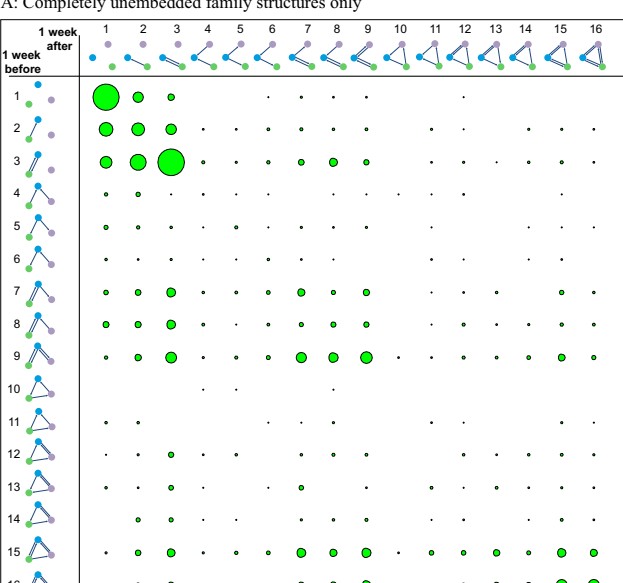
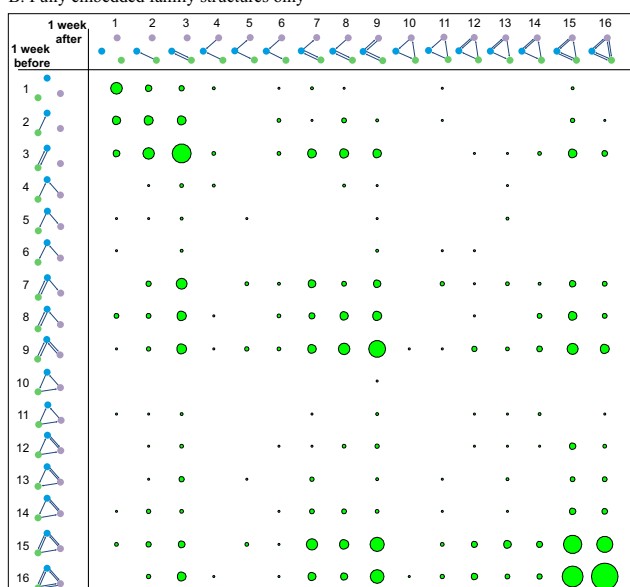

**Fig. 4 Embeddedness structure and triadic motifs before and after earthquake.** Change in family triadic motif structures (normalized) before vs. after the earthquake for completely unembedded (**A**) and fully embedded families (**B**).

To test if larger family sizes had different social dynamics in the aftermath of the earthquake, we also collected additional data for families with four members ($N = 2223$ families) and re-estimated Models 1 and 2 for these families (Supplementary Tables 19 and 20). We obtained consistent results with baseline models, suggesting that our findings apply to larger structures beyond triads.

**Model 2: friendship network activation latency**. We next tested if embeddedness structure could predict activation latency (hours) of an ego's friendship network, i.e., how long it took for an ego to contact the first alter who was not a family plan member (Table 2). During emergencies, temporal latency may reflect concern, emotional reaction, or relationship closeness. We used a truncated regression with random effects for families and applied the same independent variables as Model 1. The results were consistent with those of Model 1. A negative interaction term showed that at higher earthquake intensity levels, egos in completely unembedded family structures were significantly faster in calling friends; and a positive interaction term shows that at higher earthquake intensity levels, egos in fully embedded family structures were significantly slower in calling friends. In other words, at higher earthquake intensity, people in completely unembedded families exhibited greater urgency in reaching out to friends, while people in fully embedded families were less urgent. We obtained the same pattern of results when the dependent variable was latency until the first inbound call from a friend (Supplementary Table 6), which measured how quickly an ego's nonfamily social network came to their support.

**Temporal evolution of family motifs**. Our next two models investigated changes in social network structure and dynamics for up to 10 weeks beyond the immediate shock of the earthquake. To visualize the change in family structures after the earthquake, we graphed the relative change in family motif structures 1 week before and after the earthquake (Fig. 4 and Supplementary Figs. 13–16)—the timing of which was, of course, exogenously specified. Triadic family motifs are graphs where the directed edges denote the occurrence of directed communications between two nodes[23,31,37]. There are 16 unique types of possible motifs[38]

of mobile communications within traidic family that can be classified into three groups: one edge motifs, two edge motifs, and three edge motifs (see Supplementary Information regarding triadic motifs).

Completely unembedded families exhibited a non-preferential shift from higher to lower numbered triads after the earthquake. However, fully embedded families had more direct communications with each other both before and after the earthquake (Supplementary Figs. 13–16). Fully embedded families had more three-edged motifs (42.9–49.9%), balanced motifs (31.9–37.7%), and non-vacuous transitive motifs (22.8–28.7%) both before and after the earthquake, as compared to completely unembedded families (three-edged, 16.6–26.1%; balanced, 28.0–33.0%; non-vacuous transitive, 8.9–14.0%; Supplementary Table 25). This likely reflects that fully embedded families contain stronger dyadic ties as well.

**Model 3: communications reciprocity**. If egos in more fully embedded family structures have more responsive and supportive relationships among themselves and with friends, then they should have more reciprocal communications in the earthquake's aftermath. We counted total number of reciprocal communications across the span of our panel data (see "Methods"), where a call is reciprocal if the alter of an outbound (inbound) call was also the same alter of the next inbound (outbound) call. It should be noted that we do not distinguish communications to family or friends here. We tested if embeddedness structure qualified the exogenous impact of the earthquake on reciprocity, controlling for the fixed effects of the families (Table 3 and see "Methods"). To delineate the effects of different embeddedness structures on reciprocity during the earthquake, we contrasted individuals with fully embeddedness family structures against those with completely unembedded family structure. The outcome variables had a relatively parallel trend before the earthquake; while the earthquake increased reciprocity for both groups, the increase was noticeably larger for the fully embedded group (Supplementary Figs. 9c and 11).

Consistent with previous models, we observed a significant positive interaction between the fully embedded family dummy and the earthquake dummy. In other words, the earthquake

enhanced quantity of reciprocal communications for people with fully embedded family structures, relative to those with completely unembedded family structures. Communications frequency had significant positive main effects, suggesting that more active communicators also had more reciprocal communications. However, people with higher degree centrality (i.e., more friends) had fewer reciprocal communications, which might reflect a capacity constraint for those with more social ties. We further conducted a robustness check by controlling for the effects of individual and period fixed effects (Supplementary Table 9); the results were all consistent with those of Model 3.

**Model 4: centrality**. The results of Models 1 and 2 suggest that families with more embedded structures placed relatively greater emphasis in intra-family communications in the aftermath of the earthquake. However, it is conceivable that more embedded family structures come at a cost of relative insularity and fewer extra-family communications; after all, if one prioritizes family over friends and not the other way around, then, by definition, one's relationships with friends are relatively less important. We explored this question by testing if embeddedness structure qualified the impact of the earthquake on degree centrality, i.e., social network size, over time, which reflects people's relative sociability. Model 4 used the same statistical approach and logic as Model 3 (panel model with fixed effects, testing the interaction between embeddedness structure and earthquake), except with degree centrality as the dependent variable (Table 4 and see "Methods"). We also performed a robustness check controlling for individual and period fixed effects, and obtained the same results (Supplementary Table 10).

The significant positive interaction term shows that fully embedded families have significantly higher degree centrality after the earthquake. Since the earthquake likely increased social support needs, the positive interaction term suggests that egos in fully embedded families were able to activate and mobilize a relatively larger social support network after suffering the earthquake. An intuitive explanation is that fully embedded families by definition share more social ties (as seen from the significantly positive main effect of fully embedded structure); hence, an ego may benefit from their family alters' social networks and have a larger effective social network to call upon (and receive calls from) in the aftermath of the earthquake. This likely reflects capacity to marshal social resources and not just number of previous social ties; families with completely embedded structures also had higher degree centrality before the earthquake. Overall, this suggests that family ties can be a complement rather than substitute for nonfamily ties from an ego's friendship social network.

**Variance of family embeddedness**. Finally, we test an alternative operationalization of embeddedness structure, variance of family embeddedness (which is continuous rather than categorical), in statistical models that are analogous to and use the same dependent variables as Models 1–4 (Supplementary Fig. 12 and Supplementary Tables 15–18). These analyses show how a structural and triadic conceptualization of embeddedness offers insights that are otherwise unobservable from a purely dyadic perspective.

We define variance of family embeddedness as the variance of the number of embedded nonfamily friends that each dyad in the family triad has. This measure reflects the relative imbalance in shared social ties in a family; e.g., in a balanced structure where all three family dyads have similar numbers of embedded ties, variance is low; in an unbalanced structure where one or two dyads have many more embedded ties, variance is high. The

variance measure has the advantage of reflecting the weighting or relative differences in the strength of embeddedness (i.e., overlap parameter) of the different family dyads. It is noteworthy how this measure differs from a family's overall "embeddedness strength", i.e., mean number of embedded nonfamily ties in each dyad, which does not require a triadic or structural perspective. For example, if family A has three dyads that all have two embedded ties, and family B has two dyads with one embedded tie and one dyad with four embedded ties, then families A and B have the same embeddedness strength (two), but different variance in family embeddedness (six).

We first use the same statistical approach as nonfamily 1 and 2 to test the interaction between variance of family embeddedness and the embeddedness structure dummy. We find that egos in fully embedded family structures are even less (more) likely to call friends (family) first (Supplementary Table 15), and are relatively slower in activating their nonfamily social network (Supplementary Table 16) if their family has higher variance in embeddedness. Both results suggest that egos in fully embedded family structures are even more likely to prioritize family communications, when the families are unbalanced in embeddedness. One possible explanation is that in fully embedded family structures, imbalance signals the presence of one particularly close dyadic relationship, which received relative prioritization immediately after the earthquake.

We then use the same statistical approach as Models 3 and 4, and explore how the mean and variance of family embeddedness affect post-earthquake social dynamics. We find a significant negative interaction effect between mean and variance of family embeddedness for both reciprocity and centrality (Supplementary Tables 17 and 18). In other words, even families with more shared social resources will have relatively less reciprocity and lower degree centrality if they have an unbalanced embeddedness structure. This qualification of the impact of embeddedness strength (i.e., overlap parameter) again underscores that a triadic or higher-level perspective of embeddedness can offer deeper behavioral insights than a purely dyadic perspective.

## Discussion

Families' social dynamics after the earthquake depended, to a significant degree, on each family's triadic embeddedness structure. Despite being an indirect measure of the cohesion and overlap of individual family members' networks, the mere structure of a family's embeddedness could predict their communications patterns after a major earthquake, in terms of their communications behavior immediately following the emergency, relative orientation of social support (family vs. friends), and reciprocity dynamics over time. Embeddedness structure was a stronger predictor of communications behavior than dyadic measures of relationship strength, such as tie strength (communications frequency) and embeddedness strength (overlap parameter), which have received more attention in the empirical literature.

A triadic or higher-level perspective of embedded relations is fundamentally different from "pairwise paradigm"[29,30]. We argue that in a family triad, the embedded relationships of the other family members may also influence an ego's behavior. For example, an ego in a family where the other two alters share social resources with each other might be worse off than an ego in a completely unembedded family because the former has a more inequitable distribution of social resources.

Conceptually, families' embeddedness structure may reflect the overall balance of shared social resources within a family and the interrelationship between family members' respective friendship networks. Thus, the existence of embedded social ties between

family members denotes whether there are non (core) family alters who can potentially reinforce intra-family ties, and enmesh individual families within shared social network neighborhoods, particularly after the sudden shock of disasters. Similarly, our analyses of variance of embeddedness shows that "imbalanced" embeddedness within families, i.e., less overlap and sharing of social resources, predicted less reciprocal family communications, and lower degree centrality after the earthquake. Overall, more fully embedded families were able to both prioritize core family communications and also marshaled more nonfamily social connections. This suggests that embeddedness structure both reflects how well-integrated the whole family is with individual members' social networks, and also how effectively dyadic family relationships are reinforced by their mutual nonfamily ties. More fundamentally, more or less sharing of social resources within family resources likely corresponds to level of information sharing, homophily, trust, and emotional closeness[16–20].

However, more research is needed to explore the precise social mechanisms by which a family's shared social resources and overlapping network topographies affect their social behavior. We observed several empirical puzzles, including numerous triadic motifs with persistently unbalanced and unreciprocated relationships, i.e., family members who do not communicate with each other (Fig. 4, Supplementary Figs. 13–14, and Supplementary Table 25), which contravene the principles of structural balance theory and transitivity[26,39–42]. One possible explanation is that embeddedness allows relatively unconnected family members to be indirectly connected. For example, a fully embedded family might have friends who are better integrated into the family social network, diffuse information, and help maintain mutual ties. Similarly, a completely unembedded family may need to rely more on direct communications for relationship maintenance and information diffusion, which could explain why they had more triads that satisfy the transitive property[27].

Our research also makes methodological contributions to social networks research. Although we utilize family plan data that with naturally occurring triangle structures, future research using standard CDR datasets, where group membership is not formally defined, can potentially use embeddedness structure as an alternative means of grouping triads, detecting clusters, or testing the cohesiveness of groups (e.g., based on exogenously defined group membership). Although we argue that triads are the basic building blocks for all network structures[23,31,37–39,43], future investigations may extend and generalize our findings to higher-order structures: for example, our robustness checks (Supplementary Tables 19 and 20) extend our findings to four-person families. Future research differentiating the behavioral impact of different embeddedness structure types (and going beyond dichotomous categorical independent variables) may require finer data and models of temporal and evolutionary dynamics.

We also provide insight on how societies respond to disaster[4–9,44]. Using large-scale social network data, we show that social dynamics after a natural disaster depends, to a significant degree, on the structural features of a family's embeddedness. Families represent more than arbitrary local subgraphs; they are basic components of society. As such, family embeddedness structure also has implications on a macro-network level. Although the detailed evolutionary dynamics of the giant connected component is beyond the scope of this research, we documented its initial rapid formation from three-person networks after the earthquake (Supplementary Figs. 16–19). Overall, although intra-family calls were the dominant form of communications (363,184 calls, a 30.7% increase), there were also more inter-family calls (86,918 calls after the earthquake, a 49.2% increase from previous week). These effects were also heterogeneous across clusters. Our results suggest that differences can

be explained by differences in family embeddedness structure. Nonetheless, why some family subgraphs and meso-level clusters prioritize intra-family over inter-family calls remains an open question. Functionally, intra-family communications facilitate relationships, support, and coordination within families, while inter-family communications promote greater interconnectivity and information diffusion at the macro-network level. To the extent that family ties represent strong (as opposed to weak) ties, this may also reflect relative prioritization of strong vs. weak tie networks, each of which have different relative advantages and functions[16,45]. In turn, one may wonder how such functional roles qualify family structures' broader social response to natural disasters, and by extension, societies' social interconnectivity and resilience to system-wide shocks.

## Methods

**Telecommunications data.** We used 4 months of anonymized telecommunications records (March 1st to June 30th, 2013), regarding 35,565 active subscribers (who were in 11,855 three-person family plans) of a major Chinese mobile telecommunications carrier, who were residing in the Ya'an region of Sichuan. The data included time-stamped records of individuals' voice calls, text messages (SMS), mobile internet usage, mobility (tower access), demographics, and customer data (e.g., phone model, spending, and family plan membership).

The earthquake occurred at 08:02 on April 20th. Physical presence in Ya'an was verified by mobile phone cell tower reception during the earthquake; the dataset also identified roaming subscribers, which we use in robustness checks. The communications-based (e.g., call-derived) independent variables for Models 1 and 2 were measured continuously before the earthquake; WeChat and instant messaging app usage frequency were measured up to the time of the dependent variable, i.e., after the earthquake. For Models 3 and 4, there were observability gaps for the dependent variables (reciprocity and centrality), which were measured for one-week time spans starting −4, −1, +1, +4, and +7 weeks from the earthquake. We created five adjacent time periods of panel data by matching these observations with corresponding independent variable data from the data set that continuously spanned March 1st to June 30th. Note that a total of 16,922 individuals with either fully embeddedness family structure or completely unembedded family structure during the five periods are analyzed in the panel data fixed effects models.

We also checked the stability of the call data and the impact of WeChat adoption in China on our data; we did not find evidence of a substitution effect during our period of study. Indeed, between January 2013 and June 2016, voice call usage was stable and only text message usage declined (Supplementary Fig. 2A, B), which supports the validity of the call data in our study period. Nonetheless, for robustness, we later include WeChat and instant messaging app usage as control variables in our statistical models. Finally, the general pattern of calls over this period suggests that outside of Lunar New Year, calling patterns are relatively stable, i.e., there are no seasonality effects.

We use telecom family plan membership to proxy family membership, and note throughout the paper that our terminology "family" and "friends" are used to distinguish "family plan members" and "nonfamily plan members", respectively. Although this proxy is imperfect, it is objective, and the mobile carrier checks applicants' identity cards upon registration for family plans. Previous research typically relies on surveys to identify family relationships, which is limited by biases in accuracy, selective memory, sampling, and quantifiability. We argue that it is socially meaningful that individuals signed up for family plans together, that the primary account holder incurred the economic burden (or at least risk) of paying for others' mobile telecommunications services in a relatively low-income region of China. It is also a signal of (revealed) closeness that an individual chooses to pay for the mobile phone bill of some kin but not others.

To cross-validate the family plan data, we conducted a phone survey of all numbers in 2000 randomly drawn family plan subscriptions from the dataset 2 years after the earthquake. We called and surveyed the 6000 phone numbers individually (response rate = 45.7%; see Supplementary Information), asked participants to identify the family roles (e.g., mother, paternal grandfather, etc.) of the other users in their plan: 40.8% were parents, 31.1% were couples, 22.6% were children, 1.71% were grandparents, and 3.83% others; average family size was 3.70; 72.5% of the family plan members lived together.

**Friendship network activation model (Model 1).** The first model examined the likelihood of an ego's first outbound call after the earthquake to a friend, as opposed to family. We specified $U_{ij}$, the utility of individual $i$ in family $j$ to contact a nonfamily tie after the earthquake, with the following model:

$$U_{ij} = \mathbf{X}_{ij}\mathbf{b} + \mathbf{Z}_j\boldsymbol{\delta}_j + e_{ij} \qquad (1)$$

Where $\mathbf{X}_{ij}$ includes all explanatory variables as previously described for individual $i$ in family $j$. $\boldsymbol{\delta}_j$ are family random effects, assumed to be multivariate normal with

mean 0 and variance $\Sigma$. $\mathbf{b}$ is a vector of regression coefficients on the fixed predictors, $\mathbf{Z}_j$ is a matrix of random predictors, and $e_{ij}$ is an independent error term. Incorporating random effects, the model assumes $\mathrm{Cov}(\boldsymbol{\delta}_j, \mathbf{X}_{ij}) = 0$. We used the logit link to identify the nonfamily tie activation $A_{ij} = 1$ (activation) or 0 (non-activation) as follows:

$$\mathrm{logit}[\mathrm{Pr}(A_{ij} = 1 | \mathbf{X}_{ij}, \mathbf{Z}_j)] = U_{ij} \qquad (2)$$

Where $\mathrm{logit}[\mathrm{Pr}(\cdot)] = \log\left(\frac{\mathrm{Pr}(\cdot)}{1-\mathrm{Pr}(\cdot)}\right)$ is the logit link.

**Friendship network activation latency model (Model 2)**. Activation latency (hours) depends on an ego's latent intention of contacting a nonfamily tie; the values should range continuously from negative to positive since it is also possible to harbor strong intentions against contacting a tie (as opposed to weak intentions of contacting a tie). Since the observed activation latency is observed only when it is positive (i.e., truncated at 0), we employed a truncated regression model where the ego's latent intention of contacting the important tie is

$$y_{ij}^* = \mathbf{X}_{ij}\boldsymbol{\theta} + \mathbf{W}_j\boldsymbol{\eta}_j + \varepsilon_{ij}, \varepsilon_{ij} \sim N[0, \sigma^2], \qquad (3)$$

Where $\mathbf{X}_{ij}$ includes all explanatory variables for individual $i$ in family $j$ as in Model 1. $\boldsymbol{\eta}_j \sim \mathrm{MVN}[0, \boldsymbol{\Omega}]$ are family random effects. $\boldsymbol{\theta}$ is a vector of regression coefficients, and $\mathbf{W}_j$ is a matrix of random predictors. Using random effects, the model assumes $\mathrm{Cov}(\boldsymbol{\eta}_j, \mathbf{X}_{ij}) = 0$. Since we only observe real activation latency $y_{ij} = y_{ij}^*$ when $y_{ij}^* > 0$, the model specification adjusts for the truncation problem in the following way:

$$\mathrm{E}(y_{ij} | y_{ij} > 0) = \mathbf{X}_{ij}\boldsymbol{\theta} + \mathbf{W}_j\boldsymbol{\eta}_j + \sigma\lambda(\alpha_{ij}) \qquad (4)$$

Where $\lambda\left(\alpha_{ij}\right) = \frac{\phi(\alpha_{ij})}{1-\Phi(\alpha_{ij})}$ and $\alpha_{ij} = \frac{-\mathbf{X}_{ij}\boldsymbol{\theta}}{\sigma}$, $\phi(.)$ and $\Phi(.)$ are the standard normal density and its CDF, respectively.

Estimating the truncated model with so many random effects resulted in slow- or non-convergence of parameters. We thus used the Bayesian approach to estimate the model parameters because this approach avoids well-nigh impossible numerical integrations over high-dimensional family random effects $\boldsymbol{\eta}_j$ (i.e., 10,693 families), and empirically generates robust results with zero-truncated and zero-inflated data[46,47]. Specifically, we estimated the proposed model using Stan with appropriate and non-informative priors. The prior distribution for the parameters was set as $N(0, 5^2)$ (we also experimented with other non-informative priors, such as $N(0, 3^2)$ and $N(0, 10^2)$; the estimation results were similar). We generated four chains, each of which contained 4000 iterations with the first 2000 samples being discarded as burn-in, and started from different initial values to monitor convergence. Model diagnostics including the Rhat and effective sample sizes are reported in model results. The Rhat of parameters were closed to 1, and the ratios of the effective sample sizes over the total sample sizes of parameters were all >0.5, suggesting the HMC (Hamiltonian Monte Carlo) algorithm converged well. No divergent transitions were identified during sampling. In addition, we plotted the overlaid histograms of the (centered) marginal energy distribution $\pi_E$ and the first-differenced distribution $\pi_{\Delta E}$; the plot suggests that the momentum resampling-induced energy distributions were uniformly equal to the marginal energy distribution, further evidencing that HMC achieved optimal performance. Detailed model diagnostics including a sample of trace plots for the parameters of the two interaction terms can be found in Supplementary Fig. 10.

**Panel data fixed effects models (Model 3 and Model 4)**. To test the differential effects of earthquake intensity on the social network behavior of egos with the two different family embeddedness structures of interest (fully embedded vs. completely unembedded triadic structure), we set up a panel data fixed effects model with the following equation:

$$\begin{aligned} Y_{ijt} = {} & \beta_1 \mathrm{Embedded}_{ijt} + \beta_2 \mathrm{Earthquake}_t + \beta_3 (\mathrm{Embedded}_{ijt} \times \mathrm{Earthquake}_t) \\ & + \mathbf{C}_{ijt}\boldsymbol{\alpha} + u_j + \in_{ijt} \end{aligned} \qquad (5)$$

Where $Y_{ijt}$ is the social network outcome variable (reciprocity or degree centrality) for individual $i$ in family $j$ that is at period $t$. $\mathrm{Embedded}_{ijt}$ is a dummy variable that takes value one if individual $i$ belongs to experiment group (i.e., fully embedded triad) at period $t$. $\mathrm{Earthquake}_t$ is a dummy variable indicating whether the period is after the earthquake (i.e., equals one after period 2). $\mathbf{C}_{ijt}$ is a vector of controlling variables including ego's degree centrality (in the reciprocity model), ego's call, text, and internet frequency, phone price, whether the ego used instant messaging, was out of town during the earthquake (roaming), lives in a rural district, and whether the telecom tower was damaged during earthquake. $u_j$ are family fixed effects, controlling for potentially unobserved family-level factors. $\in_{ijt}$ is a random, idio-syncratic error term. $\beta_3$ is our coefficient of interest, examining the differential effects of fully embedded triads and completely unembedded triads (on network outcome) during earthquake. Note that by using fixed effects, the model allows the covariance of family fixed effects $u_j$, with other model covariates to be nonzero. In choosing between fixed effects and random effects of families for the panel data analysis, Hausman test results suggest that the proposed fixed effects model is

better (Supplementary Tables 9 and 10). We used R (version 3.5.1) to estimate all models in this study.

Although this analysis is not strictly a difference-in-difference analysis (since we cannot manipulate embeddedness structure), the model specifications are comparable; also, comparisons of the outcome variables between the proposed experiment (fully embedded triad) and control groups (completely unembedded triad) benefit from the exogeneity of the earthquake. The patterns for both social network outcome variables (reciprocity and centrality) were parallel before the earthquake; the occurrence of the earthquake increased the outcome variables for both groups, but the increase in magnitude was much larger for the fully embedded group (Supplementary Figs. 11 and 13A, E).

**Data protection and human subjects approval**. The data was provided by a major Chinese telecommunications carrier under a confidential agreement. All personal data were anonymized into unique ID's for analysis. Ethical approval to use anonymized individual-level telecom data for academic research is granted by the Human Research Ethics Committee of The University of Hong Kong (EA1912107).

**Reporting summary**. Further information on research design is available in the Nature Research Reporting Summary linked to this article.

## Data availability
Our contract with the telecom carrier prevents us from sharing the full telecommunications dataset publicly. Sample data and aggregated statistics for replication and academic research purposes are available from the corresponding author on reasonable request.

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

## Acknowledgements

We thank Mark Granovetter, Robert Weyer Jr., and David Rand for their feedback. We thank an unnamed national carrier in China for providing the anonymized data, and their staff for their assistance in data preparation. J.S.J. is supported by the Research Grants Council of Hong Kong (C7105-20G, 14505217, and 17506316). J.J. is supported by the National Natural Science Foundation of China (72042009 and 72074072). Y.L. is supported by the Lam Woo Research Fund at Lingnan University. X.L. is supported by the National Natural Science Foundation of China (91846301, 82041020, 72025405, and 72088101).

## Author contributions

J.S.J., J.J. and Y.L. designed research; J.S.J., J.J., Y.L., X.L., Y.N. and N.A.C. performed research; J.J. and Y.N. collected the data; Y.L., J.J., Y.N., X.L. and J.S.J. analyzed data; J.S.J. and N.A.C. wrote the paper. J.S.J., J.J. and Y.L. contributed equally to this paper.

## Competing interests

The authors declare no competing interests.

## Additional information

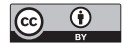

