## [Peer Review File · Nature Communications]

Reviewers' comments:

Reviewer #1 (Remarks to the Author):

The authors propose a measure of embeddedness for social groups and test how it helps to predict different types of behaviour in response to an important shock (here an earthquake).

The work is extremely well written and provides solid theoretical arguments for their empirical study. In my view, the contributions of this work are manifold, including a shift from dyadic interactions to higher-order interactions, and understanding social mechanisms underpinning social behaviour in extreme conditions.

I am overall in favour of the publication of this manuscript in Nat Comm after the following points are considered:

- The analysis relies on the family plan of a mobile phone operator. This plan imposes a structure based on triangles, but the authors could speculate on how to generalise the analysis in the case of standard CDRs.

- As noted above, the authors focus on higher-order interactions. I would recommend to make connections to recent works promoting this type of generalisation, including <https://doi.org/10.1038/s41567-019-0459-y> or the very relevant <https://doi.org/10.1073/pnas.1800683115>

- I found it difficult to find a clear and unambiguous definition of embeddedness. I would recommend the authors to provide a clear definition for instance in the caption of the relevant figure. Related to this, it was also unclear to me whether thresholds are used to decide if two people are friends or not.

- Other metrics could be used to characterise the cohesion of a triangle, such as the balance between the weights (e.g. number of calls) of its 3 links. Did the authors consider this possibility?

Reviewer #2 (Remarks to the Author):

I really enjoyed reading this paper. The data set is unique, analyses are very interesting, and the concept of family embeddedness promises to shed new light on the interaction of social network structure and social processes. The paper is well written. That said, I do have several major comments that would be good to address in a revision.

Can the authors provide more details about the demographics and family roles of families with different levels of embeddedness? They briefly mention that “40.8% were parents, 31.1% were couples, 22.6% were children, 1.71% were grandparents, and 3.83% others; average family size was 3.70; 72.5% of the family plan members lived together.” Could some of those categories be included in the analyses to better understand what determines family embeddedness – e.g. whether the family plan has children, whether they live together etc? I understand this can be done only for the part of the data but this

would nevertheless be very informative and further strengthen the conclusions.

I would avoid references to casual effects of embeddedness on social network dynamics, because without controlling for all other possible factors that could have influenced different calling patterns after earthquakes, it is hard to make causal conclusions.

I would also avoid conclusions about how close-knit families are, and stick to empirically measurable indices such as how often they call each other. The term close-knit is not well defined.

Finally, I would avoid saying that the paper explores “the process” by which earthquakes affect “social order”, as mentioned in the last paragraph of the discussion. Embeddedness is a structural property, not a process; and “social order” seems both vague and overreaching.

Minor comments:

In the survey data mentioned above, percentages sum to 100, but surely one can be a parent and a member of a couple, etc?

“Difference in difference analysis is often described as the next best thing to a natural experiment” – I’d rephrase this, especially as there is only one reference given.

Reviewer #3 (Remarks to the Author):

This paper examines the call patterns within family units in the aftermath of an emergency—an earthquake. It finds, essentially, that (controlling for prior communications) families that share many ties with third parties (i.e., are embedded) have higher density and prioritization of within family communication after the crisis. Interesting idea, but there are so many questions on the empirical analysis side that I lost confidence in the paper.

A few concerns:

1) I think the authors mischaracterize their research design. They claim that they are using difference in difference analysis. But diff in diff usually means that the treatment group gets the treatment due to some exogenous shock, and that the difference (post minus pre test) can be compared to a control group that did not get the shock. So, for example, they cite a recent paper (Caruso and Miller, 2015) examining the impact of an earthquake on human capital investments. In that case, the treatment group is the people in the areas hit by the earthquake; and the control group are people from, other,

comparable towns, that were not hit by the earthquake. The paper, then, further, examines differential effects of the earthquake across gender, finding a higher impact on women than men. The comparison, in all cases, is between people in the earthquake affected area to those not. In the current paper, however, the treatment group are highly embedded families and the control group low embedded families. Embeddedness is not affected by the earthquake, & thus the paper cannot claim the advantages of diff in diff for inferential purposes. What does this mean, practically? It means that the treatment (embeddedness) is much more likely to be correlated with other things. For example; perhaps highly embedded people live in apartment buildings; and living in apartment buildings is related to various kinds of interdependencies, both within or between families. Or, there might be any of a number of other things; the point is that you can't reasonably assert that the treatment is exogenous (that said; in many applications of diff in diff that assumption is shaky).

That said, I wouldn't fetishize the magic of diff in diff. I think the pattern of results could be interesting. But the authors need to more critically assess their causal assertions.

2) Conceptually, I am unpersuaded by the value of looking at triads. Imagine, for a moment, that the paper were redone just looking at family units size 2. What would be lost? The stat models would change, of course, but in terms of the paper's theoretical contributions? There is a discussion, for example, around impact on triadic motifs—but really, isn't the basic point of it is that the more embedded the triads, the bigger the "treatment" effect? The reason why I push on this is that the discussion around triads seems to obscure the main argument.

3) I am confused by some of the specifications. Consider the results reported in table 1. One of the independent variables is "Dyadic embeddedness with important tie." Another is "ln(Tie strength with important tie)". These two variables dominate the model (z stats of -29 and -46 !). First, any z stats that large require a careful look—perhaps partially a big N artifact, but second, what's going on here? First, it's a bit weird to use information in an independent variable that builds in information on the DV (i.e., trying to predict who the important tie will be, so using information on who the important tie is on the right hand side of the equation does not seem kosher). Second, it is not clear why the relationship is so freakishly strong—given that the important tie could be family or non family. So, basically, what these results are saying are that if you have a super strong tie to the person that you called first after the emergency, you are super likely to call your family first. That's a very problematic analysis; and the paper does nothing to unpack this. I'll note similar weirdness in other tables. E.g., in table 3, DV is reciprocity of ego with other family members, and ln(Total call frequency of ego) has a t stat of 122. I.e., if ego calls a lot then there is likely to be reciprocity within the family. I can see why that is—i.e., if someone calls a lot then they are likely to call their family a lot; but there needs to be some evaluation of a control variable that so dominates a model.

4) Lots of things are fairly undefined. For example, consider the measurement of reciprocity. Reciprocity is "the total number of reciprocal calls between an ego and his two alters." Given that every call is, technically, a directed action, this begs the question of what a reciprocal call is. So—if A calls B once and B calls A 100 times, is that 101 reciprocal calls? I am also confused by the Ns for each analysis. In reviewing now, I think that some of the specifications re-use observations across periods to get treatment effects(?). That's a big problem.

5) There are lots of SUTVA violations here; there is lots of interference (indeed, that's the point), and likely heterogeneity of treatment (the latter is partially dealt with in some of the models). I think (but

am not sure) that all 3 egos in triads are used in ego level analyses, for example. Not sure how to deal with it, but it should be discussed.

Response Letter to Reviewers

Summary of changes:

Based on the review team's feedback, and to address the issues raised by R3 in particular, we have completely overhauled the paper's statistical analyses (all statistical analyses, including those in the SI, have been re-done and expanded as appropriate). We have also tightened the exposition of our conceptualization and results throughout. We have rerun Models 1-4 (Tables 1-4 in main text) with stricter model specifications, including random effects for Models 1-2, family-level fixed effects for the panel data Models 3-4, and a new set of control variables throughout, in order to address the various statistical questions raised by the reviewer. These are accompanied by updated robustness checks (Supplementary Tables 3-9) and new robustness checks using a continuous count of dyadic embeddedness as an alternative operationalization of the key independent variable (Supplementary Tables 10-13). Finally, as another demonstration of the value of a higher-ordered triadic perspective, we conducted a new set of analyses on the impact of mean versus variance of triadic embeddedness (Models 5-8; Supplementary Figure 11, Supplementary Tables 15-18). We also add and update figures throughout the main text and SI. If anything, these additional analyses further strengthen our conclusions.

Below, we address each reviewer's comments in turn.

Reviewer 1's Comments:

Thank you for your positive and encouraging comments.

We detail our response and amendments below each specific comment:

The work is extremely well written and provides solid theoretical arguments for their empirical study. In my view, the contributions of this work are manifold, including a shift from dyadic interactions to higher-order interactions, and understanding social mechanisms underpinning social behaviour in extreme conditions.

I am overall in favour of the publication of this manuscript in Nat Comm after the following points are considered:

- The analysis relies on the family plan of a mobile phone operator. This plan imposes a structure based on triangles, but the authors could speculate on how to generalise the analysis in the case of standard CDRs.

We have added a paragraph to the general discussion speculating about additional applications for standard CDR datasets (with no family identification); these include using triadic embeddedness structures as an alternative means of grouping triads or detecting clusters. For network datasets with group membership data (e.g., organizational hierarchies, co-authorship

networks, school networks etc.), triadic embeddedness structure can potentially be used as an alternative test of the cohesiveness of those exogenously defined triads and groups.

However, we also believe our conceptualization can be used generally (i.e., even when triads are arbitrarily defined or identified). Triads are the smallest form of a network, are present in all datasets, and have long been the subject of research on network structures (at least since Simmel's original work in the 1890's!). As you point out, a key message of our research is that additional insights may be derived when analyses shifts from the examination of inter-relationship within triads (e.g., balance and transitivity) to the study of the structural properties of a triad's relationships with alters outside the triad.

- As noted above, the authors focus on higher-order interactions. I would recommend to make connections to recent works promoting this type of generalisation,

Thank you for your recommendation. We have cited these two recent works and explicitly state in our literature review that our work intends to contribute to a growing body of research showing the advantages of studying higher-order interactions and topographies in networks.

- I found it difficult to find a clear and unambiguous definition of embeddedness. I would recommend the authors to provide a clear definition for instance in the caption of the relevant figure. Related to this, it was also unclear to me whether thresholds are used to decide if two people are friends or not.

We now provide a clearer exposition of the theoretical concept of structural embeddedness in the front end of the paper, while also detailing the concept's history and typical operationalization in the literature (largely from a dyadic perspective based on measures such as the "overlap parameter" – which is a count of the number of friends [C, D, E.....] that any two people, A and B, have in common). We also provide a more detailed and precise definition of "structure of triadic embeddedness", our primary construct of interest, and provide the definition in the relevant figure.

Our definition and operationalization of structural embeddedness is based on a dummy variable (there is a link if two nodes share at least one common friend; there is no link if two nodes do not share any common friends). We thus do not use any thresholds. We have edited the paper to make this clearer.

However, we do explore the possibility of threshold effects by conducting additional robustness checks using a numeric measure of the mean of triadic family embeddedness (i.e., overlap parameter of the entire family; Supplementary Tables 11-14) in lieu of the family embeddedness structure dummy variable used in our primary analyses. The purpose of this robustness check is

to show that the analyses in the main text, which uses a dummy variable for family structure, hold even when we use a continuous measurement to operationalize family embeddedness. This alleviates concerns that our main results were a mere result of the dichotomous operationalization of embeddedness (or if there is a latent threshold effect). All results, and the signs of the interaction terms, are consistent with the main results that use the dichotomous/dummy operationalization.

- Other metrics could be used to characterise the cohesion of a triangle, such as the balance between the weights (e.g. number of calls) of its 3 links. Did the authors consider this possibility?

Thank you for this interesting suggestion. To explore the idea of balance, we do something a bit different that uses the number of calls, but we develop this suggestion and conduct a new set analyses using the variance of embeddedness as the independent variable of interest. Specifically, we measure variance of the number of embedded non-family friends that each dyad (i.e., link between two nodes) in the family triad has. The variance measure reflects weighting or relative differences in the strength of embeddedness of the different family dyads. In addition, it is empirically tractable and easily measured; we hope our empirical analysis of the construct may be useful for future researchers.

We discuss the results in the main text and SI (starting pg. 27 in SI; Supplementary Figure 11; Supplementary Tables 15-18). In Models 5-6, we find that variance interacts with family embeddedness structure. In particular, egos in fully embedded family structures are even more likely to prioritize family communications when their families are *unbalanced* in embeddedness. One possible explanation is that in fully embedded family structures, imbalance signals the presence of one particularly close dyadic relationship, which received relative prioritization immediately after the earthquake.

In Models 7-8, we show that variance of embeddedness has an independent effect and also interactive effect with mean of embeddedness (i.e., overlap parameter of whole family). We find that a family triad with more shared social resources will still have less reciprocity and lower degree centrality if it is unbalanced in embeddedness structure. This qualification of the benefit of greater embeddedness strength (i.e., overlap parameter) again underscores that a triadic, structural perspective of embeddedness can offer deeper behavioural insights than a purely dyadic operationalization of embeddedness.

Overall we believe our analyses and the exposition of the new balance construct you have suggested helps highlight that moving from dyadic to triadic (and general higher order)

conceptualizations of embeddedness structure can yield new behavioural and conceptual insights. Again, we are grateful for all of R1's suggestions.

Reviewer 2's Comments:

We are grateful for your encouraging comments as well as clear suggestions. We address to each comment in turn:

I really enjoyed reading this paper. The data set is unique, analyses are very interesting, and the concept of family embeddedness promises to shed new light on the interaction of social network structure and social processes. The paper is well written. That said, I do have several major comments that would be good to address in a revision. Can the authors provide more details about the demographics and family roles of families with different levels of embeddedness? They briefly mention that "40.8% were parents, 31.1% were couples, 22.6% were children, 1.71% were grandparents, and 3.83% others; average family size was 3.70; 72.5% of the family plan members lived together." Could some of those categories be included in the analyses to better understand what determines family embeddedness – e.g. whether the family plan has children, whether they live together etc? I understand this can be done only for the part of the data but this would nevertheless be very informative and further strengthen the conclusions.

In response to this good suggestion, we now provide additional tables detailing the average demographics and family roles in each embeddedness structure type (Supplementary Tables 3-4). We do not find significant differences between the groups. However, we urge caution in interpreting these results or conducting formal analyses using the survey data because of data limitations (both inherent methodological constraints of a telephone survey, and because fully embedded structures were over-represented in the sample). We provide more details of on pg. 8 of the SI. We would like to reiterate that the primary purpose of the survey data was to provide a sanity check (that family plan members were typically kin) and basic intuition about the composition of typical active families.

Still, we do however believe that R2's intuition that family role (and relationships within the family triads) likely plays a role in determining family social dynamics and embeddedness structure; however, we may need longitudinal/panel data (survey or otherwise) to study this issue.

I would avoid references to casual effects of embeddedness on social network dynamics, because without controlling for all other possible factors that could have influenced different calling patterns after earthquakes, it is hard to make causal conclusions.

We have adopted more conservative wording in the interpretation of our statistical models.

I would also avoid conclusions about how close-knit families are, and stick to empirically measurable indices such as how often they call each other. The term close-knit is not well defined.

We have removed conclusions about “close-knitted-ness” and have removed the term throughout, including from our title. We have rephrased relevant claims around measurable behaviours as you have suggested.

Finally, I would avoid saying that the paper explores “the process” by which earthquakes affect “social order”, as mentioned in the last paragraph of the discussion. Embeddedness is a structural property, not a process; and “social order” seems both vague and overreaching.

We have edited the paper accordingly. Thank you.

Minor comments:

In the survey data mentioned above, percentages sum to 100, but surely one can be a parent and a member of a couple, etc?

This is true and a good catch. The results you mention sum to 100 because they were coded to reflect the ‘primary relationship role’ every individual had in their families (which the interviewer asked). We have made note of this in the relevant SI section (Family Survey, pg. 8).

“Difference in difference analysis is often described as the next best thing to a natural experiment” – I’d rephrase this, especially as there is only one reference given.

We have removed this claim and that specific reference. We have also updated the writing and the Methods that we use “Panel Data Fixed Effects Models” with earthquake occurrence as an interaction term. We discuss these changes in more detail in our reply to Reviewer 3.

Overall, we thank R2 for the very helpful suggestions, all of which we have endeavoured to implement.

Reviewer 3’s Comments:

We are grateful for your detailed comments and critical feedback, particularly in regard to the statistical analyses. We have revamped all of the statistical models, and replaced every statistical

table in the old main text and SI with new analyses. We detail the changes below and how they address the concerns you raise. We believe the paper and our results have become much stronger as a result of the changes.

We summarize the changes we have made below before responding to your comments in turn:

1. For the two “quasi-DID” models (which we now refer to as panel models), we control for family-level fixed effects, and (very robustly) clustered the standard errors of the parameters at the family level (i.e. we address the issue that the treatment is on family level, while the outcomes are individual-level, by addressing unobserved family confounding factors and delineate the effect sizes (of parameters) with the consideration of family-level robust standard errors);

2. For the other two tie activation/latency variables, due to the cross-sectional rather than panel nature of the data, we used the random effects instead of fixed effects. A completely re-written Methods section provides the details of each model;

3. The stricter model specifications (listed as points 1 and 2 above) also make the t and Z values for the controlling variables much smaller than before. We have also calculated the VIFs (all smaller than 4) for the controlling variables of the models, so the relatively large t or Z values for certain controlling variables (e.g. calls) are not problematic/influential, and model results remained similar both in significance and magnitude as we excluded those controlling variables with large t or Z values;

4. The differences in N (number of observations for Tables 1 and 2 versus Tables 3 and 4) are due to differences in the data frames of the two set of analysis. Specifically, the data for the tie activation/latency analysis is cross-sectional while the data for the “quasi-DID” analysis was panel data. Due to data constraints (in how much data the carrier could provide us), there were observability gaps for the dependent variables of the panel data models (i.e., reciprocity and centrality). We note this more clearly in our revision (e.g., in Methods) and address your questions directly below;

5. We have deleted the two potentially problematic controlling variables (dyadic embeddedness and tie strength with the important tie, as they are directly correlated with the DVs as reviewer 3 suggested), and the results do not change meaningfully;

6. We now control for time-variant WeChat usage (i.e., we control for WeChat usage frequency, which replaces the original dummy variable of having WeChat app), and further controlled for the usage frequency of other communication apps which may serve as alternative communication channels. However, we emphasize in the SI that communications apps were unlikely to have

significantly substituted voice calls even under non-emergency contexts during the year of the study.

Minor notes:

In Tables 3 and 4, the main effect of the variable "Ego and family all embedded" is perfectly collinear with family fixed effects, and thus was excluded from model estimation. This does not affect the model's main purpose, which is to test for the interaction between embeddedness structure and earthquake.

Additional notes about the statistical models can now be found in the footnotes of each table.

R3's comments:

This paper examines the call patterns within family units in the aftermath of an emergency—an earthquake. It finds, essentially, that (controlling for prior communications) families that share many ties with third parties (i.e., are embedded) have higher density and prioritization of within family communication after the crisis.

Clarifications: We study more than prioritization (which was Models 1-2); Models 3-4 study reciprocity and degree centrality (for both family and non-kin). In addition, the overarching point of the paper is to say more than simply what makes a close family or triad (which is one goal), but to, quote R1 and R2, showcase how higher-level network structures (and attendant conceptualizations) can be used to understand relationships and behaviours in networks.

1) I think the authors mischaracterize their research design. They claim that they are using difference in difference analysis. But diff in diff usually means that the treatment group gets the treatment due to some exogenous shock, and that the difference (post minus pre test) can be compared to a control group that did not get the shock. So, for example, they cite a recent paper (Caruso and Miller, 2015) examining the impact of an earthquake on human capital investments. In that case, the treatment group is the people in the areas hit by the earthquake; and the control group are people from, other, comparable towns, that were not hit by the earthquake. The paper, then, further, examines differential effects of the earthquake across gender, finding a higher impact on women than men. The comparison, in all cases, is between people in the earthquake affected area to those not. In the current paper, however, the treatment group are highly embedded families and the control group low embedded families. Embeddedness is not affected by the earthquake, & thus the paper cannot claim the advantages of diff in diff for inferential purposes. What does this mean, practically? It means that the treatment (embeddedness) is much more likely to be correlated with other things. For example; perhaps highly embedded people live in apartment buildings; and living in apartment buildings is related to various kinds of interdependencies, both within or between families. Or, there might be any of a number of other things; the point is that you can't reasonably assert that the

treatment is exogenous (that said; in many applications of diff in diff that assumption is shaky).

That said, I wouldn't fetishize the magic of diff in diff. I think the pattern of results could be interesting. But the authors need to more critically assess their causal assertions.

Thank you; we quite agree. We now more properly describe the model as “Panel Data Fixed Effects Models.” Moreover, to strengthen the robustness of our statistical inferences, we have:

(a) controlled for family fixed effects (of more than 6000 families) and clustered the standard errors (of the parameters) at the family level for all panel data analysis, which addresses the concern that family embeddedness (structures) are correlated to interdependencies within families (e.g., family members living in the same buildings are likely highly embedded);

(b) We have also added two new control variables "Total WeChat frequency" and "Total frequency of other instant messaging" to try to mitigate certain concerns about interdependencies between families, for example, families living close to each other can have interactions through WeChat and other instant messaging apps;

(c) Finally, even though our framework is indeed not a DID analysis (since, as R3 has suggested, embeddedness is not affected by the earthquake), the earthquake is still an exogenous shock. We try to build this logic in our statistical analyses of how family embeddedness structure can predict social communications behaviours that to a great extent stem from exogenous factors like earthquake risk.

2) Conceptually, I am unpersuaded by the value of looking at triads. Imagine, for a moment, that the paper were redone just looking at family units size 2. What would be lost? The stat models would change, of course, but in terms of the paper's theoretical contributions? There is a discussion, for example, around impact on triadic motifs—but really, isn't the basic point of it is that the more embedded the triads, the bigger the “treatment” effect? The reason why I push on this is that the discussion around triads seems to obscure the main argument.

In the revised main text, we clarify the theoretical contributions and empirical benefits of studying “higher order interactions” (to borrow R1's phrase) such as triads. The general argument that a triadic perspective can provide additional insight is not new, and has received extensive theoretical and empirical treatment in the social networks and sociology literature (we provide a clearer review in the 4th paragraph). As R1 points out, there is also a growing body of recent work demonstrating the advantages of studying higher order interactions and network topographies. From a purely informational perspective, shifting from a dyadic to triadic perspective reveals more information and a more nuanced understanding of relationship context. Indeed, there are numerous important theoretical constructs that cannot be studied dyadically, e.g., structural balance, transitivity, motifs, structural holes, etc.

The goal of our paper (and our empirical analyses) is to show that the mere structure of a group's embeddedness structure (as expressed by links that denote the presence of embedded relationships) can predict numerous meaningful social network and behavioural outcomes.

Indeed, our motivation for including controls such as dyadic tie strength and embeddedness strength in our original main tables was to demonstrate that the additional effect of higher level structure above and beyond traditional dyadic measures.

Nonetheless, we agree it is our responsibility to demonstrate how a triadic level of analysis may bring additional conceptual advantages over using just dyadic measurements of embeddedness. To address this question empirically, we conduct a new set analyses (Supplementary Tables 15-18) that use the mean of family embeddedness (which is essentially based on the dyadic overlap parameter extended to 3 people) and variance of family embeddedness (variance of the number of embedded non-family friends that each *dyad* of the triad has) as the key independent variables. We show that the two have independent effects, and that the variance measure provides new insights (e.g., by allowing for the study of structural balance, which is a theoretical construct that by definition cannot be studied dyadically; please see our comments in response to R1 on pg. 3).

3) I am confused by some of the specifications. Consider the results reported in table 1. One of the independent variables is “Dyadic embeddedness with important tie.” Another is “ln(Tie strength with important tie)”. These two variables dominate the model (z stats of -29 and -46 !). First, any z stats that large require a careful look—perhaps partially a big N artifact, but second, what’s going on here? First, it’s a bit weird to use information in an independent variable that builds in information on the DV (i.e., trying to predict who the important tie will be, so using information on who the important tie is on the right hand side of the equation does not seem kosher). Second, it is not clear why the relationship is so freakishly strong—given that the important tie could be family or non family. So, basically, what these results are saying are that if you have a super strong tie to the person that you called first after the emergency, you are super likely to call your family first. That’s a very problematic analysis; and the paper does nothing to unpack this. I’ll note similar weirdness in other tables. E.g., in table 3, DV is reciprocity of ego with other family members, and ln(Total call frequency of ego) has a t stat of 122. I.e., if ego calls a lot then there is likely to be reciprocity within the family. I can see why that is—i.e., if someone calls a lot then they are likely to call their family a lot; but there needs to be some evaluation of a control variable that so dominates a model.

Thank you for these detailed comments and perceptive suggestions. In response, we have (a) excluded “Dyadic embeddedness with important tie” and “ln(Tie strength with important tie)” (which are highly correlated with the DVs) from the models in this revision’s main text. The model results (for the key covariates and interactions), as shown in this revision, remained similar both in magnitude and significance after deleting these two covariates. And (b), we have also checked the VIFs of all the covariates (including the two covariates with large Z stats in Tables 1 and 2) in all original models under a linear setting; the VIFs of all covariates are smaller than 4, suggesting the two dominating variables, as well as ln(Total call frequency of ego) in Tables 3 and 4 do not have an undue effect on the parameter estimations of the other variables. Finally, (c), we have controlled for family fixed effects and, in particular, clustered the standard

errors (of the parameters) at family levels; subsequently the t value for $\ln(\text{Total call frequency of ego})$ in Tables 3 and 4 has reduced substantially from 122/121 to 38/43. Again, we thank R3 for these careful comments, which have prompted us to implement better modelling.

4) Lots of things are fairly undefined. For example, consider the measurement of reciprocity. Reciprocity is “the total number of reciprocal calls between an ego and his two alters.” Given that every call is, technically, a directed action, this begs the question of what a reciprocal call is. So—if A calls B once and B calls A 100 times, is that 101 reciprocal calls? I am also confused by the Ns for each analysis. In reviewing now, I think that some of the specifications re-use observations across periods to get treatment effects(?). That’s a big problem.

We provide a clearer definition of reciprocal calls earlier in the main text. We conservatively define communications as reciprocal if the alter of an outbound (inbound) call is also the same alter of the next inbound (outbound) call (i.e., if the person calling the ego is also the person the ego calls next, and vice versa).

In the Methods section, we clarify the number of observations used. The change of N (number of observations) is due to the different data frames of the two sets of analysis: panel data fixed effect analysis for the DVs of reciprocity and centrality (16,922 distinct individuals across 5 periods*), and the cross-sectional data of the tie activation/latency analysis (26,191 distinct individuals for just one tie-activation period). In other words, the number of subjects/individuals involved in each analysis vary due to different model setups, for example, not-fully-embedded families are included in the tie activation/latency analysis but not in panel data fixed effect analysis in which we only kept fully embedded and fully unembedded groups respectively as quasi-experiment and quasi-control groups.

*note that in order to provide a clear contrast between individuals with fully embedded family structure and completely unembedded family structure, observations of individuals whose family embeddedness structure changed from either fully embeddedness family structure or completely unembedded family structure into not-fully-embedded family structure in certain periods were further excluded from the analysis (the results are robust even without this exclusion).

5) There are lots of SUTVA violations here; there is lots of interference (indeed, that’s the point), and likely heterogeneity of treatment (the latter is partially dealt with in some of the models). I think (but am not sure) that all 3 egos in triads are used in ego level analyses, for example. Not sure how to deal with it, but it should be discussed.

The most challenging SUTVA issue, as you suggest, is possibility of interference between family-level embeddedness and ego-level analysis (i.e., all 3 egos in triads are used in ego level analyses). To tackle this potential spillover between units at the family level and ego level, (a) for the panel data analysis (on reciprocity and centrality), we explicitly control for family-level fixed effects (for more than 6000 families), and (robustly) clustered the standard errors of the parameters at the family level, therefore substantially ruling out unobserved family confounding factors that influence our observed effects; (b) for the other two models on tie activation and

latency, due to the cross-sectional rather than panel nature of the data, we instead control for the random effects of ~20,000 families, also addressing the SUTVA concerns of within-family spillovers.

In closing, we would like to reiterate our gratitude for the review team's clear feedback, which has helped us to prepare a much stronger revised manuscript. We appreciate your time and effort in handling our original submission and this new revision.

Many Thanks!

Sincerely,
Jayson S. Jia

On behalf of Jianmin Jia, Yiwei Li, Xin Lu, and Nicholas A. Christakis

Reviewers' comments:

Reviewer #1 (Remarks to the Author):

The authors have addressed my minor comments. I am thus in favour of the publication of this manuscript in Nature Communications.

Reviewer #2 (Remarks to the Author):

The authors have thoroughly revised many aspects of the paper. These revisions have in my view adequately addressed reviewers' concerns. I believe this will be an important and well-cited paper and am recommending it for publication.

Reviewer #3 (Remarks to the Author):

Regarding triads—since this is fundamental to the thesis of the paper: the fact that family plans have three people seems immaterial to the general argument. The relevant triads, in a Simmelian sense, should be the triads of family pairs with “non-family” third parties. The fact that the paper focuses just on families of 3 and ignore other sized families really does not shed any light on triads. Note that the identical statistical analyses could have been done if family plans had been for 2 people, 4 people, or an indefinite number of people. And, putting the stats aside, conceptually, by the logic of the paper, wouldn't we expect that families of 5 that were embedded (i.e., shared lots of third party connections) would prioritize each other? (If not, why not?)

Indeed, the focus on threesomes renders the analyses somewhat confusing. Ego could be completely non-embedded with their two family members (zero overlapping nonfamily friends), but not counted as unembedded because their two family members are embedded. What is the justification for this?? Indeed, why aren't the analyses at the intra-family dyad level, since intra-family dyads vary in their embeddedness? Wouldn't we expected embedded intra-family ties to be stronger, and prioritized during the earthquake? Further, if the analyses were whether dyads were commonly embedded with third parties, then there would be a nice continuous measure of embeddedness. The categorical classification of families as fully embedded/partially embedded/not embedded is an odd coding decision, given that embeddedness should be viewed roughly as a continuous (really count) variable, if measured at a dyadic level: do A and B have 0, 1, 2, 3... friends in common.

Per Granovetter, we would expect any dyad (family or not) to be a stronger tie (anytime) if it were embedded. The argument does not apply only to family relationships. I would expect that generally, more deeply embedded ties are stronger, and that it just happens that family ties are more likely to be strong, and also be highly embedded. Also, I'd note that there are likely family members who are not part of family plans (i.e., "family plan" does not equal "family" and that many of the "non-family" ties

may be family). The analysis conducted in this way cannot distinguish the difference between impact of kinship versus dyadic strength versus embeddedness.

The survey numbers, as reported in the main paper, are misleading. As the SM notes, they only surveyed members of family plans who actively (at least 10 intra-family calls) called each other after the earthquake. This is not representative of their sample, since presumably some (unknown fraction) of their sample did not have that level of activity. A smaller point, but also note that this survey includes families of various sizes (average size was 3.7), so does not represent the sample used in the paper. Finally, would have been better to report the frequency of different family types—since the survey captured the report regarding the entire family. Thus, reporting just the relationship to the respondent is a bit weird. Family 1, with a parent and grandparent (relative to the respondent) is really just the same as family 2, with child and grandchild. Although I may be understanding the data incorrectly. This sentence regarding the survey is confusing to me:

“*note that these results sum to 100 because they were coded to reflect the ‘primary relationship role’ every individual had in their families. More generally, individuals may have multiple ‘roles’ within families.”

I assume, except in rare cases, that everyone in a family has a single relationship to the respondent. So, if I am reading this right—does this mean that the mother of the respondent can be coded in the data as “mother” (of respondent), “daughter” (of the grandfather of the respondent, who is also in the family plan), and “wife” (of the father who, also happens to be in this family plan)? What I would propose would be to plot 3 person, isomorphic family trees (A and B are parents to C is the same family structure as A with parents B and C).

Small point: is the fact that there are no reported siblings in the data just a reflection of the one child policy? (Assuming so, but checking.) Were there any nonfamily members reported?

Let’s imagine the results just reflect the fact that “embedded families of 3” are really just family plans that cut across households of connected extended families. So: ego calls someone on their family plan because they prioritize family, and this alter lives in a different household (think adult child with their own spouse calling their parents). Is this an interesting result? Or is it just that CDRs are correlated with an omitted factor that is driving the pattern of behavior observed?

What is the incentives around utilizing family plans (as compared to other types of plans)—do they greatly reduce per person costs? Or is there some maximum threshold where they are allowed? Or where the incentive disappears? This will affect the incorporation of (extended) family into family plans.

There should be some reporting of the substantive size of the effects—and not just the treatment effects, but of the control variables.

It is hard for me to wrap my mind around the motivation behind the inclusion of these particular control

variables. Caution is of particular importance, because many of these variables are endogenous, and are, by construction, likely to be related to the RHS variables of interest, creating a high risk of biasing the parameter estimates of the impact of the focal variables. Further, some of these controls just dominate the regressions—t stats of 38 and 42 (!), greatly amplifying the risk that they are substantially affecting the other parameter estimates.

If the argument is that the earthquake changes behavior to prioritize family members in embedded families, shouldn't the analyses control for prior share of intra-family calls? (and if that is not the argument, what analytic weight is the earthquake carrying here?)

Wouldn't we expect a high proportion of cohabitating families to be together, physically, at 8am on a Saturday? (and thus, presumably, not calling each other right after the earthquake?) This would dramatically affect what our expectations are. This is culturally embedded, so some guidance here is necessary. Did the survey ask which of these family members the respondent was with during the earthquake?

Models 3 and 4 use fixed effects instead of the random effects used in Models 1 and 2. Can you discuss the decision to use fixed effects here, perhaps including the results of a Hausman (1978) test in the SM? Actually, could you be more explicit about what you mean in both sets of models by fixed and random effects? There is significant variability in what these terms mean from discipline to discipline

The details around the estimation of the Bayesian model are inadequate. The sampling procedure is not described, nor are the priors on the parameters being estimated. The only evaluation of the sampling of the model is the inclusion of R^2 in the regression tables; while R^2 is a useful diagnostic it alone is insufficient to show that sampling of the model was adequate. See <https://arxiv.org/abs/1903.08008> for a discussion of R^2 's limitations. I'm not suggesting the SM necessarily needs to have page after page of trace plots, but some discussion of (i) priors, (ii) software used for sampling, and (iii) model diagnostics beyond R^2 would be appropriate. For example, if the model was estimated using Stan, what are the effective sample sizes for the parameters. Were there divergent transitions during sampling?

I believe from one use of the word "previous" that the Internet use, WeChat use, etc. variables are all restricted to the pre-earthquake period, but if this could be made explicit in the discussion of model specification that would be helpful.

Figure 3 is difficult to interpret and adds little to the argument.

I believe from one use of the word "previous" that the Internet use, WeChat use, etc. variables are all restricted to the pre-earthquake period, but if this could be made explicit in the discussion of model specification that would be helpful.

Reviewer 1

The authors have addressed my minor comments. I am thus in favour of the publication of this manuscript in Nature Communications.

We thank R1 for these positive comments. The previous changes have improved the comprehensiveness and clarity of our paper, and we are grateful.

Reviewer 2

The authors have thoroughly revised many aspects of the paper. These revisions have in my view adequately addressed reviewers' concerns. I believe this will be an important and well cited paper and am recommending it for publication.

We are grateful to R2 for these encouraging comments. We believe that the structural aspects of embeddedness remain understudied in the literature despite being a key tenet of social network theory.

Reviewer 3

We are grateful to R3 for taking the time to review our paper. We have endeavored to address all of R3's points and new requests, and believe our paper has improved as a result. However, we note that we actually discussed or addressed many of the conceptual issues raised in the original manuscript (pg. 3-4, 6, 12-14).

Regarding triads—since this is fundamental to the thesis of the paper: the fact that family plans have three people seems immaterial to the general argument. The relevant triads, in a Simmelian sense, should be the triads of family pairs with “non-family” third parties. The fact that the paper focuses just on families of 3 and ignore other sized families really does not shed any light on triads. Note that the identical statistical analyses could have been done if family plans had been for 2 people, 4 people, or an indefinite number of people. And, putting the stats aside, conceptually, by the logic of the paper, wouldn't we expect that families of 5 that were embedded (i.e., shared lots of third party connections) would prioritize each other? (If not, why not?)

Our paper studies family social network dynamics in a triadic context in terms of *embeddedness* structure (and importantly how such embeddedness structure influences social behavior after a major natural disaster). “Triads of family pairs with ‘non-family’ third parties” is not the focus of this paper. But because we study *embeddedness* structure and since “edges” in the triad reflect embeddedness (rather than tie strength), we are in a sense actually studying how family triads link to outside “non-family third parties”, and how this affects real social behavior.

We disagree that studying family triads in terms of embeddedness does not contribute to the literature on triads and social networks. We have already shown that the structure of triads (operationalized by either structure or mean vs. variance of triadic embeddedness) has an impact on social network behaviour beyond dyadic embeddedness (which we controlled for). We also

choose to study family triads since three person families are the modal family structure in China, which besides being representative, is in itself a pertinent topic of study.

Nonetheless, to address R3's concerns and new requests, we have extended our analyses to larger families in this revision. We have collected new data from the carrier for families with 4 members (2223 families in total) and re-estimated our benchmark models (see Supplementary Tables 19-20). The results were consistent with that of the benchmark models in both signs and significance, suggesting that our findings are applicable beyond families of three members. i.e., an ego whose family is more embedded (regardless if the family has 3 or 4 members) is generally less likely to call non-family members after an earthquake, and also takes longer time to do so.

Indeed, the focus on threesomes renders the analyses somewhat confusing. Ego could be completely non-embedded with their two family members (zero overlapping nonfamily friends), but not counted as unembedded because their two family members are embedded. What is the justification for this?? Indeed, why aren't the analyses at the intra-family dyad level, since intra-family dyads vary in their embeddedness? Wouldn't we expected embedded intra-family ties to be stronger, and prioritized during the earthquake? Further, if the analyses were whether dyads were commonly embedded with third parties, then there would be a nice continuous measure of embeddedness. The categorical classification of families as fully embedded/partially embedded/not embedded is an odd coding decision, given that embeddedness should be viewed roughly as a continuous (really count) variable, if measured at a dyadic level: do A and B have 0, 1, 2, 3... friends in common.

We note that the literature has thoroughly studied dyadic relations and embeddedness¹. The entire raison d'être of this paper is to move beyond the dyad and study higher-order interactions and topographies in social networks. We have since the first manuscript explained in the introduction how the structural role of triads is understudied despite being theoretically important as the smallest possible social network structure²⁻⁷.

There are several reasons why our primary analyses use a categorical classification of family structures and not a continuous variable:

- (1) The most important reason is that different family embeddedness structures are not equivalent, as R3 suggests. Even conceptually, an ego unembedded within a family where everyone is unembedded is different from an ego who is unembedded from an embedded dyad. The point here is that the relationship of the other family members may also influence the ego's behavior. For example, it is possible that parents who have a strong rather than a weak relationship with each other (e.g., are embedded or not) are more likely to influence their adult child. Alternatively, an ego in a family where the other two alters share social resources with each other might be worse off than an ego in a completely unembedded family because the former has a more inequitable distribution of social resources.
- (2) We test the specific hypothetical that R3 brought up with a new robustness check in the SI (Supplementary Tables 21-22). Specifically, we directly compare the differential impact of these two family structures (with the fully unembedded group as the reference). The results suggest that when ego is unembedded but alters are embedded, the ego was less likely to call non-family members after an earthquake, and took longer to do so (as compared to when ego and alters were all unembedded). This shows that the two family

structures are different in their social response to the earthquake. However, a dyadic coding scheme would have overlooked this effect since it does not consider how the embeddedness of the alters in a triad affects an ego.

- (3) Using a categorical (dummy) coding scheme, as proposed by R3, to study family embeddedness structure would generate similar results as a dyadic continuous coding of “mean and variance of family embeddedness” (i.e., mean of overlap parameters of all dyads in a family). We have already done this as a robustness check. This measure yields consistent results as the categorical classification of family structures (see SI Tables 11-14).
- (4) The data for categorization dummy variables are easier to collect and less intrusive with regards to data privacy. For example, we show that it is not necessary to measure exactly how many friends family members have in common. This may facilitate analysis by scientists, governments, or NGO’s studying the social impact of sudden disasters.

Per Granovetter, we would expect any dyad (family or not) to be a stronger tie (anytime) if it were embedded. The argument does not apply only to family relationships. I would expect that generally, more deeply embedded ties are stronger, and that it just happens that family ties are more likely to be strong, and also be highly embedded. Also, I'd note that there are likely family members who are not part of family plans (i.e., "family plan" does not equal "family" and that many of the "non-family" ties may be family). The analysis conducted in this way cannot distinguish the difference between impact of kinship versus dyadic strength versus embeddedness.

R3 raises two issues; (1) the relationship between embeddedness and tie strength (for family and friendship ties), and (2) whether the data identifies kinship relationships. We address each in turn:

(1) It should be noted that family ties (as defined in our paper) are not always the strongest ties, as measured by either tie strength before the earthquake or ‘revealed importance’. In our data, the majority of the individuals first called non-family rather than family members after the earthquake, suggesting that non-family relationships can also be important and strong.

Indeed, in the initial version of our paper, we actually included ego’s dyadic tie strength and dyadic embeddedness with the important tie as control variables (in Tables 1 and 2) to address and pre-empt these issues. We removed those control variables (“Dyadic embeddedness with important tie” and “ln(Tie strength with important tie)”) in the previous revision round at R3’s request (because they are highly correlated with the DVs). We attach the results below for reference (Tables 1 and 2) since they address the issue R3 now raises.

In this revision, we have also included “Prior intra-family outbound call share” as a control variable in all our models to address your concern that “family ties are more likely to be strong, and also be highly embedded” (we discuss inclusion of this control variable later in this letter).

Overall, our analyses (with different variations of control variables included) all yield consistent results with our main results, and we do not find evidence of the issues R3 raises.

Dependent Variable = p (first outbound call is to non-family plan member)

Coef.	S.E.	z	P> z
-------	------	---	------

(1) Ego and family all unembedded	0.548	0.071	7.76	<0.001	***
(2) Ego and family all embedded	-0.292	0.029	-10.00	<0.001	***
Earthquake intensity group (1 = severe)	0.089	0.037	2.38	0.017	*
(1)*earthquake intensity group	0.241	0.110	2.19	0.028	*
(2)*earthquake intensity group	-0.011	0.045	-0.24	0.814	
ln(Degree centrality of ego)	0.056	0.026	2.16	0.030	*
ln(Total call frequency of ego)	0.222	0.020	11.26	<0.001	***
ln(Total text frequency of ego)	0.012	0.007	1.67	0.095	.
ln(Internet usage frequency of ego)	-0.009	0.006	-1.34	0.179	
ln(Phone retail price in Yuan)	-0.005	0.014	-0.38	0.707	
ln(Total Wechat frequency)	0.007	0.008	0.88	0.381	
ln(Total frequency of other instant messaging)	0.003	0.007	0.51	0.611	
Roaming dummy (1 = traveling outside of prefecture)	-0.357	0.028	-12.58	<0.001	***
Rural dummy (1 = rural)	-0.081	0.022	-3.62	<0.001	***
Damage dummy (1 = cell towers damaged in neighborhood)	0.255	0.076	3.37	<0.001	***
Prior intra-family outbound call share	-1.534	0.078	-19.67	<0.001	***
Dyadic embeddedness with important tie	-0.371	0.011	-34.86	<0.001	***
Tie strength with important tie	-0.451	0.012	-36.88	<0.001	***
(Intercept)	-0.176	0.098	-1.79	0.074	.

Note. Other settings were the same as in Table 1

Table 1: Robustness Check for Estimates in Table 1 of Body Text (including dyadic embeddedness and tie strength with important tie)

Dependent Variable = Time until first outbound call to non-family plan member (hours)	Coef.	S.D.	Lower 2.5% CI	Upper 97.5% CI	N_eff	Rhat
(1) Ego and family all unembedded	-30.300	4.862	-40.127	-20.882	18359	1.000
(2) Ego and family all embedded	68.534	4.544	59.677	77.350	14300	1.000
Earthquake intensity group (1 = severe)	-15.582	4.533	-24.413	-6.727	15401	1.000
(1)*earthquake intensity group	-16.390	5.138	-26.618	-6.121	14954	1.000
(2)*earthquake intensity group	16.669	4.712	7.349	26.037	15067	1.000
ln(Degree centrality of ego)	-68.620	4.340	-76.980	-59.905	11911	1.000
ln(Total call frequency of ego)	-56.003	3.594	-63.085	-48.949	10133	1.000
ln(Total text frequency of ego)	-26.170	2.812	-31.685	-20.625	12587	1.000
ln(Internet usage frequency of ego)	1.997	2.529	-2.979	6.973	11414	1.000
ln(Phone retail price in Yuan)	35.716	2.672	30.461	41.002	9731	1.000
ln(Total Wechat frequency)	-7.436	3.034	-13.367	-1.531	12637	1.000
ln(Total frequency of other instant messaging)	-13.495	2.721	-18.821	-8.084	10539	1.000
Roaming dummy (1 = traveling outside of prefecture)	26.171	4.793	16.838	35.554	18109	1.000
Rural dummy (1 = rural)	21.499	4.435	12.768	30.071	15591	1.000
Damage dummy (1 = cell towers damaged in neighborhood)	-1.421	4.940	-11.209	8.271	14668	1.000
Prior intra-family outbound call share	28.984	4.964	19.246	38.881	15356	1.000
Dyadic embeddedness with important tie	126.818	2.966	121.025	132.635	11855	1.000

Tie strength with important tie	138.834	3.003	133.010	144.767	14861	1.000
(Intercept)	12.418	4.896	2.867	22.098	13846	1.000

Note. Other settings were the same as in Table 1

Table 2: Robustness Check for Estimates in Table 2 of Body Text (including dyadic embeddedness and tie strength with important tie)

(2) For CDR data from telecom data, telecom family plan is the most objective way of measuring family membership. Previous research typically relies exclusively on surveys to identify relationships between families, which has its own limitations. Here, the telecom company is supposed to check applicants' identity cards upon registration for family plans. Furthermore, it is economically meaningful that the individuals signed up for family plans together, and that the primary account holder incurred the burden (or at least risk) of paying for others' mobile telecommunications services in a relatively low income region of China. Similarly, it is socially meaningful (and akin to revealed closeness) that an individual chooses to pay for the mobile phone bill of some kin but not others.

The phone survey on family relations also provides a validity test/evidence that these were predominantly family relationships.

We do not claim that our data is perfect and clearly state throughout the paper that our terminology 'family' and 'friends' are used to distinguish 'family plan members' and 'non family plan members', respectively. Nonetheless, we believe our study (and the data) is a first in many regards, especially in studying family social dynamics at scale and using verifiable data after the occurrence of a natural disaster.

The survey numbers, as reported in the main paper, are misleading. As the SM notes, they only surveyed members of family plans who actively (at least 10 intra-family calls) called each other after the earthquake. This is not representative of their sample, since presumably some (unknown fraction) of their sample did not have that level of activity. A smaller point, but also note that this survey includes families of various sizes (average size was 3.7), so does not represent the sample used in the paper. Finally, would have been better to report the frequency of different family types—since the survey captured the report regarding the entire family. Thus, reporting just the relationship to the respondent is a bit weird. Family 1, with a parent and grandparent (relative to the respondent) is really just the same as family 2, with child and grandchild. Although I may be understanding the data incorrectly. This sentence regarding the survey is confusing to me:

*“*note that these results sum to 100 because they were coded to reflect the ‘primary relationship role’ every individual had in their families. More generally, individuals may have multiple ‘roles’ within families.”*

The primary purpose of the survey was to provide a check that family plan members were kin. The advantage of asking people to identify the social roles of the family plan members (to self-identified university researchers) is that it is more likely to get an honest response than asking 'Are the members of your family plan actually your family members?'

The survey is not intended to be representative of the relative frequency of family roles (we acknowledged this immediately below the tables; see text below Supplementary Table 4). We provided information about family roles, etc. in the interests of transparency and as

additional descriptive data that we did not use in statistical analyses. As you mentioned, individuals may have multiple ‘roles’ within families; for instance, a woman can be a mother (for her child) and a daughter (if her mother also in the plan). On the other hand, our statistical analyses focused on the structure of embeddedness (and not specific family roles).

With regards to basic validity, the threshold of 10 intra-family calls is likely on the low end. We do not know total/average call frequency after the earthquake; however, before the earthquake, each ego made an average of 299 calls a month, 33 of which were to other family plan members (i.e., 1.1 intra-family calls per day per ego). On the other hand, our threshold of 1 intra-family call per day per family (i.e., 0.27 per day per ego) was 4X lower. This was conservative considering that people on average made more phone calls after the earthquake, particularly in the first 10 days.

Furthermore, since 59.5% of the carrier’s family plans (i.e., the population we sampled) were 3 person plans. In our new robustness check, we also collected data for 4-member family plans (11.1%). We obtained consistent results.

I assume, except in rare cases, that everyone in a family has a single relationship to the respondent. So, if I am reading this right—does this mean that the mother of the respondent can be coded in the data as “mother” (of respondent), “daughter” (of the grandfather of the respondent, who is also in the family plan), and “wife” (of the father who, also happens to be in this family plan)? What I would propose would be to plot 3 person, isomorphic family trees (A and B are parents to C is the same family structure as A with parents B and C).

Small point: is the fact that there are no reported siblings in the data just a reflection of the one child policy? (Assuming so, but checking.) Were there any nonfamily members reported?

Let’s imagine the results just reflect the fact that “embedded families of 3” are really just family plans that cut across households of connected extended families. So: ego calls someone on their family plan because they prioritize family, and this alter lives in a different household (think adult child with their own spouse calling their parents). Is this an interesting result? Or is it just that CDRs are correlated with an omitted factor that is driving the pattern of behavior observed?

First, as previously discussed, it’s not likely that “‘embedded families of 3’ are really just family plans that cut across households of connected extended families”, because family plans can comprise any number of family members (e.g., we conduct a robustness check for 4-member family plans).

Second, we have (even in the initial submission) ruled out the role of geographic position (i.e. different living places, traveling, etc.) with robustness checks* with roaming users only (e.g., SI Table 7). Even when the ego was out of town during the earthquake, we observed the same pattern of results as our benchmark models, e.g., an ego whose family was fully unembedded was more likely to call a non-family tie, even when they were geographically far away from the family and the family experienced severe earthquake intensity.

*We discussed these robustness checks every previous version of the main text.

What is the incentives around utilizing family plans (as compared to other types of plans)—do they greatly reduce per person costs? Or is there some maximum threshold where they are allowed? Or where the incentive disappears? This will affect the incorporation of (extended) family into family plans.

The incentive for utilizing family plans is that any family member included in the plan can enjoy price discounts. There are no threshold effects, but people may run out of family members they want to include. However, people should have no incentive to pay for other regular friends. As our survey showed, majority of members in family plan are family members from each other. Also, as previously mentioned, we have added analyses using data from 4 member family plans and obtained consistent effects.

*There should be some reporting of the substantive size of the effects—and not just the treatment effects, but of the control variables. It is hard for me to wrap my mind around the motivation behind the inclusion of these particular control variables. Caution is of particular importance, because many of these variables are endogenous, and are, by construction, likely to be related to the RHS variables of interest, creating a high risk of biasing the parameter estimates of the impact of the focal variables. Further, some of these controls just dominate the regressions—*t* stats of 38 and 42 (!), greatly amplifying the risk that they are substantially affecting the other parameter estimates.*

Firstly, calls and text messages are telecom carriers’ most basic services and widely used in models studying human social behavior⁸⁻¹⁰: we can hardly ignore them. Conceptually, we need to explicitly control for these variables (even though the *t* statistics for call frequency were 38 and 42 in Tables 3 and 4) if we want to examine the effects of the focal variables;

Secondly, as shown in the following table, the VIFs of the variables in the corresponding models were small (smaller than 4). Empirically it is not likely that the control of call (and also other controls) affected the other parameter estimates;

Variable	VIFs (Table 3)	VIFs (Table 4)
Earthquake dummy (1 = post-quake)	1.265	1.253
Ego and family all embedded * Earthquake dummy	1.421	1.377
ln(Degree centrality of ego)	3.763	NA
ln(Total call frequency of ego)	3.597	1.440
ln(Total text frequency of ego)	1.374	1.364
ln(Internet usage frequency of ego)	2.472	2.467
ln(Phone retail price in Yuan)	1.618	1.597
ln(Total Wechat frequency)	1.718	1.718
ln(Total frequency of other instant messaging)	2.489	2.489
Roaming dummy (1 = traveling outside of prefecture)	1.067	1.046
Rural dummy (1 = rural)	1.064	1.058
Damage dummy (1 = cell towers damaged in neighborhood)	1.026	1.025
Prior intra-family outbound call share	1.257	1.083

Table 3: VIFs for Variables in Table 3 and Table 4 of Body Text

Thirdly, even when we excluded the call frequency control variable from the model, the results were consistent with that of our benchmark models, as shown in the two robustness checks below.

Dependent Variable = Reciprocity between ego and two alters	Coef.	Cluster S.E.	t	P> t	
Ego and family all embedded	NA	NA	NA	NA	
Earthquake dummy (1 = post-quake)	0.323	0.041	7.92	<0.001	***
Ego and family all embedded * Earthquake dummy	0.248	0.098	2.54	0.011	*
ln(Degree centrality of ego)	0.850	0.039	21.94	<0.001	***
ln(Total text frequency of ego)	0.245	0.028	8.76	<0.001	***
ln(Internet usage frequency of ego)	0.020	0.007	3.02	0.003	**
ln(Phone retail price in Yuan)	0.159	0.041	3.90	<0.001	***
ln(Total Wechat frequency)	-0.015	0.021	-0.74	0.461	
ln(Total frequency of other instant messaging)	-0.007	0.016	-0.45	0.654	
Roaming dummy (1 = traveling outside of prefecture)	-0.149	0.079	-1.89	0.058	
Rural dummy (1 = rural)	0.096	0.207	0.46	0.643	
Damage dummy (1 = cell towers damaged in neighborhood)	0.094	0.164	0.57	0.568	
Prior intra-family outbound call share	2.197	0.198	11.07	<0.001	***
Family fixed effects	Yes				

Table 4: Robustness Check for Estimates in Table 3 of Body Text (not including call frequency)

Dependent Variable = Ego's centrality	Coef.	Cluster S.E.	t	P> t	
Ego and family all embedded	NA	NA	NA	NA	
Earthquake dummy (1 = post-quake)	0.197	0.021	9.38	<0.001	***
Ego and family all embedded * Earthquake dummy	0.217	0.057	3.82	<0.001	***
ln(Total text frequency of ego)	0.868	0.047	18.47	<0.001	***
ln(Internet usage frequency of ego)	0.037	0.009	4.22	0.000	***
ln(Phone retail price in Yuan)	0.704	0.060	11.80	<0.001	***
ln(Total Wechat frequency)	0.134	0.037	3.59	<0.001	***
ln(Total frequency of other instant messaging)	-0.055	0.025	-2.22	0.027	*
Roaming dummy (1 = traveling outside of prefecture)	-1.627	0.107	-15.19	<0.001	***
Rural dummy (1 = rural)	0.170	0.221	0.77	0.444	
Damage dummy (1 = cell towers damaged in neighborhood)	0.864	1.054	0.82	0.412	
Prior intra-family outbound call share	-2.860	0.219	-13.05	<0.001	***
Family fixed effects	Yes				

Notes. Other settings were the same as in Table 4

Table 5: Robustness Check for Estimates in Table 4 of Main Study (not including call frequency)

Lastly, the new control variable that we have just added to all models (prior intra-family outbound call share, which we will discuss next), following your suggestion, also has a very large/dominating z value in Table 1 (DV = if first outbound call is to non-family plan member), -25, see Table 1 in Main Text). We believe this is reasonable; it is not surprising that prior intra-family outbound call share affects whether the ego calls family members during the earthquake, just as it is also reasonable that ego's calling frequency influences an ego's eventual centrality and reciprocity behavior. The *t*-statistics are not problematic as long as the estimation of other parameters were not influenced.

In this revision, we have explained under the corresponding tables, that the VIFs were small and the estimation results were robust after deleting the control of call frequency, to alleviate readers' concerns that the results were biased because of the dominating controls.

If the argument is that the earthquake changes behavior to prioritize family members in embedded families, shouldn't the analyses control for prior share of intra-family calls? (and if that is not the argument, what analytic weight is the earthquake carrying here?)

Thank you for suggesting this control variable. We agree it is important and have collected the data necessary to implement this analysis. Specifically, we measured ego's call frequency to family members over their total call frequency in four weeks prior to the earthquake. We added this control in all the models and updated estimation results *of all models* (in both the main text and SI). The effects of prior share of intra-family calls were mostly significant, while the estimation results of our focal variables were basically unchanged in terms of sign and significance.

Wouldn't we expect a high proportion of cohabitating families to be together, physically, at 8am on a Saturday? (and thus, presumably, not calling each other right after the earthquake?) This would dramatically affect what our expectations are. This is culturally embedded, so some guidance here is necessary. Did the survey ask which of these family members the respondent was with during the earthquake?

As discussed earlier, the cohabitation concern is also related to the geographic position of the family members; we address this with our roaming-only sample analyses for Models 1 and 2 (which examine behavior immediately after the earthquake). It should also be noted that family members living together might still have called each other for social coordination reasons; in fact, prior research has shown that greater physical proximity and contact, for example from shared residence, often results in more, not less, mobile phone communications¹¹⁻¹³.

Models 3 and 4 use fixed effects instead of the random effects used in Models 1 and 2. Can you discuss the decision to use fixed effects here, perhaps including the results of a Hausman (1978) test in the SM? Actually, could you be more explicit about what you mean in both sets of models by fixed and random effects? There is significant variability in what these terms mean from discipline to discipline

Model 3 and Model 4 leverage the panel data structure, while Model 1 and Model 2 are based on cross-sectional data, largely because the dependent variable in Model 1 and Model 2 relates to the activation of important ties immediately after the earthquake. As you suggested, we have included the results of Hausman test in Model 3 and Model 4 to differentiate between fixed effects model and random effects model in the panel data. In our case, fixed effects were preferred in Model 3 and Model 4 as the null hypothesis were rejected in both models (For Model 3: $\chi^2 = 315.02$, $df = 5$, $p < .001$; For model 4: $\chi^2 = 751.50$, $df = 4$, $p < .001$).

By using random effects, Model 1 and Model 2 assume that the covariance of the (random) effects of families with the model covariates were 0, while fixed effects models (Model 3 and Model 4) allow the (fixed) effects of families to be correlated with the covariates. We have clarified this distinction in the Methods section and also included the results of Hausman Tests in Table S9 and Table S10 in the SI.

The details around the estimation of the Bayesian model are inadequate. The sampling procedure is not described, nor are the priors on the parameters being estimated. The only evaluation of the sampling of the model is the inclusion of Rhat in the regression tables; while Rhat is a useful diagnostic it alone is insufficient to show that sampling of the model was adequate. See <https://arxiv.org/abs/1903.08008> for a discussion of Rhat's limitations. I'm not suggesting the SM necessarily needs to have page after page of trace plots, but some discussion of (i) priors, (ii) software used for sampling, and (iii) model diagnostics beyond Rhat would be appropriate. For example, if the model was estimated using Stan, what are the effective sample sizes for the parameters. Were there divergent transitions during sampling?

Thank you for your suggestion. In this revision, we report the Bayesian estimation in detail. We estimated the proposed model using Stan with appropriate and non-informative priors. Specifically, the prior distribution for all the parameters were set as $N(0, 5)$ (we have experimented with some other non-informative priors such as $N(0, 3)$ and $N(0, 10)$, the estimation results were similar). We have generated 4 chains, each of which contained 4,000 iterations with the first 2,000 samples discarded as burn-in and started from different initial values to monitor convergence. As you suggested, besides reporting Rhat, we also report the effective sample sizes for the parameters; the ratios of the effective sample sizes over the total sample sizes of parameters were all larger than 0.5, suggesting the HMC (Hamiltonian Monte Carlo) algorithm converged well. No divergent transitions were identified during sampling. To avoid pages of trace plots, we have selectively presented the trace plots for the two interactions for readers' reference. Finally, we have plotted the overlaid histograms of the (centered) marginal energy distribution π_E and the first-differenced distribution $\pi_{\Delta E}$, which suggest that the momentum resampling-induced energy distributions were uniformly equal to the marginal energy distribution, further providing evidence that HMC achieved optimal performance. We have added detailed explanation and related model diagnostics in the model results, Methods section, and the SI.

For your convenience, we include figures Model Diagnostics for Bayesian Estimation of Model 2 below:

In SI as **Supplementary Figure 10**. Model 2 model diagnostics: **A** and **B** are trace plots for the parameters of the two interaction terms; **(C)** The ratio of the effective sample sizes over the total sample sizes of parameters; **(D)** Overlaid histograms of the (centered) marginal energy distribution πE and the first-differenced distribution $\pi \Delta E$.

I believe from one use of the word “previous” that the Internet use, WeChat use, etc. variables are all restricted to the pre-earthquake period, but if this could be made explicit in the discussion of model specification that would be helpful.

We have clarified the writing explaining this. In Models 1 and 2 (cross sectional data), the two variables "WeChat" and "Other instant messaging app" are from up to the same time the dependent variable was measured, i.e., after the earthquake. The other variables (e.g., calling behavior) are from the pre-earthquake period.

In Models 3-4, the independent variables (if time-dependent, e.g., call, text, and Internet) are all time-stamped and measured in each time period (5 periods during -4, -1, +1, +4, and +7 weeks from the earthquake). We note in the Methods section: “We created 5 adjacent time-

periods of panel data by matching these observations with corresponding independent variable data from the data set that continuously spanned March 1st to June 30th.”

Figure 3 is difficult to interpret and adds little to the argument.

Figure 3 provides a visualization of triadic motifs as well as their transition matrix. Triadic motifs have previously received extensive theoretical attention in the literature and are considered the basic building blocks of networks (e.g., Harary et al. 1965, Milo et al. 2004, Kovanen et al. 2013, Holland and Leinhardt 1970, 1976). Of course, we are open to removing the figure based on editorial feedback, or because of space constraints.

References

1. Onnela, J.P. et al. Structure and tie strengths in mobile communication networks. *Proc Natl Acad Sci USA* **104**(18): 7332-7336 (2007).
2. Milo, R. et al. Network motifs: Simple building blocks of complex networks. *Science* **298**, 824–827 (2002).
3. Milo, R. et al. Super families of evolved and designed networks. *Science* **303**, 1538–1542 (2004).
4. Kovanen, L., Kaski, K., Kertész, J., Saramäki, J. Temporal motifs reveal homophily, gender-specific patterns, and group talk in call sequences. *Proc Natl Acad Sci USA* **110**, 18070-18075 (2013).
5. Holland, P., Leinhardt, S. A Method for Detecting Structure in Sociometric Data. *Am J Sociol* **76**, 492-513 (1970).
6. Holland, P., Leinhardt, S. Local Structure in Social Networks. *Sociol Methodol* **7**, 1-45 (1976).
7. Harary F., Norman, R. Z., Cartwright D. Structural Models: An Introduction to the Theory of Directed Graphs. Wiley, New York (1965).
8. Blumenstock, J., Cadamuro, G., & On, R. (2015). Predicting poverty and wealth from mobile phone metadata. *Science*, 350(6264), 1073-1076.
9. Jia, J. S., Jia, J., Hsee, C. K., & Shiv, B. (2017). The role of hedonic behavior in reducing perceived risk: evidence from postearthquake mobile-app data. *Psychological science*, 28(1), 23-35.
10. Pokhriyal, N., & Jacques, D. C. (2017). Combining disparate data sources for improved poverty prediction and mapping. *Proceedings of the National Academy of Sciences*, 114(46), E9783-E9792.
11. Blumenstock, J. E, Eagle, N., Fafchamps, M. Airtime transfers and mobile communications: Evidence in the aftermath of natural disasters. *J Dev Econ* **120**, 157-181(2016).
12. Eagle, N., Pentland, A., Lazer, D. Inferring friendship network structure using mobile phone data. *Proc Natl Acad Sci USA* **106**, 15274-15278 (2009).
13. Saramäkia, J., Leichtb, E. A., Lópezb, E., Roberts, S. G. B., Reed-Tsochasb, F., Dunbar, R. I. M. Persistence of social signatures in human communication. *Proc Natl Acad Sci USA* **111**, 942–947 (2014).

REVIEWER COMMENTS

Reviewer #3 (Remarks to the Author):

This paper has come a long way since its first iteration, and at its core is looking at something very interesting. However, there are still serious issues, ranging from theoretical conceptualization of structural embeddedness, to measurement, to research design.

On the triad issue, the authors are wrong, but this is not as central to their findings as they argue, it's a narrative issue. The idea of structural embeddedness is simply that a relationship between two people is affected by their relationship with third parties. E.g., see Feld (1997):

The amount of structural embeddedness of a tie between two individuals is defined as the extent of overlap of social relations between those two individuals, and presumably reflects the extent of shared foci of activity that bring these individuals together with the same others.

The triad in the classic definition of structural embeddedness is that dyad + the third party(ies). That is, how many triads is a given dyad in? If the paper were looking at families of two people, their argument still holds—because a structurally embedded pair is one in which the pair shares many friends. I'll note also that operationalizations of structural embeddedness, and closely related constructs, are continuous in nature—because the amount of shared third party ties can vary. If A and B have one shared friend versus twenty, that should matter. And if A and B have one shared friend, and each has 100 other friends (versus 2), that should also matter.

Here, essentially, what the paper does is evaluate whether each dyad in the family is embedded, thresholding at 1 shared friend for each dyad (i.e., it doesn't matter if it is one shared friend or twenty), and then summing. If it sums to 0, then it's type 1; if it sums to 2, then it's type 2, etc; and then in the statistical analyses takes types 2 and 3 together as the reference category in analyses with dummy variables for type 1 and type 4 families.

Extending the notion of embeddedness to small groups (like a family) is fine—maybe great— but it is a definitional leap. The paper does offer an operationalization, but not a motivation for the operationalization, other than a vague assertion re the insights that can be garnered by looking at 3 person network motifs. So, the paper needs to acknowledge that, and work a bit more in terms of considering different ways of operationalizing, and linking that to the phenomenon in question, and deal with the fact that people within the group will be unevenly structurally embedded in the group. This conceptual leap then requires an evaluation of why we should expect the group level variable to matter rather than individual (positional) level variable to matter. This then requires a multi-level analysis. It also requires more justification of why this is a categorical variable, rather than continuous; e.g., it should matter if everyone in the family has one weak tie to a nonfamily member versus many strong ties. That measurement decision is a mystery to me. That said, tables 11-14 are reassuring. But, it's an odd choice, like dichotomizing age by saying everyone older than 60 is old, and everyone younger is

young; you might still get useful findings, but you'd be throwing away information.

Focusing on 3 member families is fine—e.g., it is the modal size family, and it usefully eliminates one element of variation in the data. But this does raise that multi-level issue, then—since in groups larger than 2, there may be variation in how structurally embedded individuals are within the group. This issue is partially addressed in Supplementary Tables 21-22 (but, ideally would be done within a multi-level model framework); but the conceptual issue is not.

I guess the biggest question here: what do we think is going on within families that we would expect structural embeddedness to matter? The standard argument in the econ soc literature is that structural embeddedness matters in markets because of concern regarding reputation among third parties. If we have many friends in common, I'm less likely to take advantage of you, because of the reputational ripple effects. But the authors really do not offer much in the way of a theory as to why structural embeddedness would matter for this particular case, other than a general argument that embeddedness matters, and then for intra-family communication, referencing Bott's classic study (which is great, but does not offer insight into why we'd expect embeddedness to matter here).

The authors write:

We used the quasi-experimental impact of the earthquake to test whether embeddedness structure and other social network constructs can affect social network dynamics in the aftermath of disaster.

Small point, but better to not write "quasi-experimental impact"—that would be like writing "RCT impact", it just doesn't make sense. It is sensible to say that the earthquake is exogenous, and the severity of the earthquake locally is unrelated to the networks of affected individuals, and that the paper takes advantage of that exogenous variation. A bigger point: it would be recommended for the paper to briefly discuss research design; what type of quasi-experimental design is it? In the first iteration of the paper, it asserted that it was using a diff-in-diff design, which it was not. Generally the paper is using the earthquake in two ways. The first is that the earthquake is a dramatic change to the context, and allows evaluation of how prior social structure relates to communication in a moment of collective crisis. The second is that the severity of the earthquake varies exogenously, and that that exogenous variation provides local variation in context. So, good to briefly unpack that.

Re family members, roaming, etc; I guess the question here is: if these families are cohabiting, and the earthquake hit them when they were home, what were they calling each other about? I.e., I assume that one would expect very different behavioral patterns if this were in the middle of a workday, when people were physically separated. This is a very specific microcontext; it's not clear that any of the referenced literature applies. It is possible, for example, that some people left their homes to go get help, and then had to call their family. Why this might be more likely for an embedded 3 person family than nonembedded is a puzzle.

Response to the reviewers' comments

Reviewers 1 and 2

We are grateful to R1 and R2 for their positive and constructive feedback in the previous rounds. Our changes based on their feedback have undoubtedly improved the comprehensiveness of our paper.

Reviewer 3

We thank R3 for taking the time to further review our paper. Our paper has benefited from critical revisions made in response to R3's comments. However, we disagree with R3's latest criticisms, particularly since they revisit many of the same issues which were raised, and subsequently addressed or clarified, in the previous two revisions. Many comments reveal a fundamental misunderstanding of our conceptual contribution and research goals. Nevertheless, we are appreciative that R3's comments helped us identify which aspects of the manuscript could be clarified and improved. We have subsequently rewritten several parts of the paper. Below, we respond to R3's comments in turn:

This paper has come a long way since its first iteration, and at its core is looking at something very interesting. However, there are still serious issues, ranging from theoretical conceptualization of structural embeddedness, to measurement, to research design.

*On the triad issue, the authors are wrong, but this is not as central to their findings as they argue, it's a narrative issue. The idea of **structural embeddedness** is simply that a relationship between two people is affected by their relationship with third parties. E.g., see Feld (1997):*

The amount of structural embeddedness of a tie between two individuals is defined as the extent of overlap of social relations between those two individuals, and presumably reflects the extent of shared foci of activity that bring these individuals together with the same others.

*The triad in the classic definition of **structural embeddedness** is that dyad + the third party(ies). That is, how many triads is a given dyad in? If the paper were looking at families of two people, their argument still holds—because a structurally embedded pair is one in which the pair shares many friends.*

R3 seems to misunderstand the goal of our paper and our concept of triadic embeddedness. R3 once again narrowly defines embeddedness from a dyadic perspective and insists on using frequency based operationalizations: We addressed these issues both using empirical evidence and conceptual reasoning in previous rounds, which R3 largely ignores. We are very aware of the predominant dyadic perspective of “structural embeddedness”; indeed, our paper's conceptual contribution and raison d'être is expanding beyond this incumbent dyadic and frequency-based empirical framework to conceptualize embeddedness using a higher order (triadic) and purely structural perspective (*discussed in next section of letter*). We have made significant changes in the writing to clarify these points and enhance framing of our contribution.

R3 suggests that embeddedness can always be reduced to a relationship between two people (and implies that a triadic or higher level perspective is unnecessary). This as overly

reductive; such an assumption would stifle future research. There are many reasons to study embeddedness from a triadic perspective (as opposed to groupings of dyads or just dyads):

- (1) The traditional concept of “structural embeddedness” is, as R3 points out, defined in a dyadic relation. However our concept of embeddedness structure is established in for a triadic motif. We use the term “the structure of embeddedness” (or “embeddedness structure”) to differentiate our key construct from “structural embeddedness”. We show that structure of embeddedness has an impact on behavior beyond the sum of the dyadic embeddedness effects. We expressly control for dyadic embeddedness strength in our analyses, and show that the structure of embeddedness has strong and statistically significant effects on social network behavior.
- (2) Triadic motifs have already received extensive theoretical attention in the literature and are considered the basic building blocks of networks (Milo et al. 2002, 2004, Kovanen et al. 2013). More recent research on networks has shown the merit of moving beyond the study of two node links and towards higher order interactions and topographies (See Benson et al. 2018 for a review). Such a shift in order and perspective produces fundamentally different insight and allows for studying higher order dependencies, modelling more complex link relationships and dynamics, making stronger causal inferences, and even decomposing otherwise unobservable indirect relationships (Benson et al. 2018, Lambiotte, Rosvall , and Scholtes 2019). Ultimately, higher order structural analyses can observe many indirect effects of how social networks affect dyadic ties that “pairwise representation” paradigms cannot.
- (3) We have previously explained why reducing analysis to pairs of ties is not the same as considering 3 or more pairs of ties concurrently. In the previous letter, we noted that different embeddedness structures might represent qualitatively different social contexts and dynamics:
“Even conceptually, an ego unembedded within a family where everyone is unembedded is different from an ego who is unembedded from an embedded dyad. The point here is that the relationship of the other family members may also influence the ego’s behavior. For example, it is possible that parents who have a strong rather than a weak relationship with each other (e.g., are embedded or not) are more likely to influence their adult child. Alternatively, an ego in a family where the other two alters share social resources with each other might be worse off than an ego in a completely unembedded family because the former has a more inequitable distribution of social resources.”
- (4) We ran a robustness expressly to provide empirical evidence for this point in the previous revision (Supplementary Table 21-22):
“We directly compare the differential impact of these two family structures (with the fully unembedded group as the reference). The results suggest that when ego is unembedded but alters are embedded, the ego was less likely to call non-family members after an earthquake, and took longer to do so (as compared to when ego and alters were all unembedded). This shows that the two family structures are different in their social response to the earthquake. However, a dyadic coding scheme would have overlooked this effect since it does not consider how the embeddedness of the alters in a triad affects an ego.”
- (5) Of course we do not claim to have complete understanding of embeddedness structure effects. Future research may more precisely quantify the differential impact of different embeddedness structural arrays (i.e., going beyond contrasting fully embedded and

completely non-embedded structures). This is, however, beyond the scope of our research, which is intended to demonstrate that such purely structural effects exist using field data.

- (6) At R3's behest, in the previous revision we also showed that structure of embeddedness effects extend to 4-person families (i.e., a higher level and for an even more complex social unit). See Supplementary Tables 19-20).
- (7) We also contribute to the literature by offering an alternative means of considering dynamics within pre-defined social units such as families (that also happen to be quite fundamental for social organization and human evolution).

We emphasize and clarify all the above points in the revised manuscript.

*I'll note also that operationalizations of **structural embeddedness**, and closely related constructs, are continuous in nature—because the amount of shared third party ties can vary. If A and B have one shared friend versus twenty, that should matter. And if A and B have one shared friend, and each has 100 other friends (versus 2), that should also matter.*

Here, essentially, what the paper does is evaluate whether each dyad in the family is embedded, thresholding at 1 shared friend for each dyad (i.e., it doesn't matter if it is one shared friend or twenty), and then summing. If it sums to 0, then it's type 1; if it sums to 2, then it's type 2, etc; and then in the statistical analyses takes types 2 and 3 together as the reference category in analyses with dummy variables for type 1 and type 4 families.

These comments miss the point of our research, i.e., we study triadic embeddedness structure effects that go beyond dyadic embeddedness effects. As per our response to the same points the last round, there are several reasons why we use a categorical classification of family structures and not a continuous variable:

- (1) The most important reason is that different family embeddedness structures are not equivalent, as R3 suggests. [Explained in (3) to (5), in pg. 3 of letter]
- (2) We provide evidence of convergent validity: Robustness checks using continuous variables to operationalize dyadic versus triadic embeddedness (i.e., mean versus variance of family embeddedness) generated consistent results (see section starting pg. 30 in SI, Supplementary Tables 15-18).
- (3) We observe effects of embeddedness structure even after controlling for the effect of continuous measures of dyadic embeddedness (I.e., overlap parameter), which we can include as a control variable.

*As we reminded R3 in the last revision letter, Tables 1 and 2 of our initial submission included overlap parameter as a control variable; we removed it in the second revision at R3's request. We now include these in the SI (Tables 23-24) as robustness checks.

- (4) We have also noted that the categorical/dummy variables are easier to collect and less intrusive from a data privacy standpoint. We also show that it is not necessary to measure exactly how many friends family members have in common to obtain consistent effects. This makes our research more useful for scientists, governments, or NGO's studying the social impact of sudden disasters.

R3's example about a dyad having "100 other friends" mattering also misses the point. We are not trying to claim that strength of dyadic embeddedness does not matter. A comparison more germane to our research interests would be: Is there a difference for post-earthquake family communication dynamics if A and B (who have 100 other friends) are in a 1) family where C

also shares friends with A and B, versus 2) a family where C does not share friends with either of them. The answer, as we show, is yes.

While we agree that studying ‘tradeoffs’ (between dyadic strength and triadic structural cohesion) or magnitude/threshold effects is an interesting topic for future study, this goes demonstrating and validating the impact of triadic embeddedness structure (as a categorical variable), which is the focus of this paper.

Extending the notion of embeddedness to small groups (like a family) is fine—maybe great— but it is a definitional leap. The paper does offer an operationalization, but not a motivation for the operationalization, other than a vague assertion re the insights that can be garnered by looking at 3 person network motifs.

The point is not merely to ‘extend the notion of embeddedness to small groups’. Our biggest contribution is offering a novel way of thinking about embeddedness (a classic topic of theoretical importance). Our conceptualization is significantly different from conventional operationalizations of embeddedness, namely dyadic overlap or contextual dummies (e.g., group membership).

Moreover, we also study of family dynamics, which is in itself an important topic (and which we study using far more objective data than previous studies, which were predominantly survey-based). See pg. 3 of this letter for a summary of our conceptual contributions.

We do not claim to have a full understanding of the micro-phenomenon underpinning our effects; but we believe we provide a thorough conceptual and empirical exposition that future researchers can build upon or extend to different domains.

We have made some minor edits to make these points even clearer.

So, the paper needs to acknowledge that, and work a bit more in terms of considering different ways of operationalizing, and linking that to the phenomenon in question, and deal with the fact that people within the group will be unevenly structurally embedded in the group. This conceptual leap then requires an evaluation of why we should expect the group level variable to matter rather than individual (positional) level variable to matter. This then requires a multi-level analysis. It also requires more justification of why this is a categorical variable, rather than continuous; e.g., it should matter if everyone in the family has one weak tie to a nonfamily member versus many strong ties. That measurement decision is a mystery to me. That said, tables 11-14 are reassuring. But, it’s an odd choice, like dichotomizing age by saying everyone older than 60 is old, and everyone younger is young; you might still get useful findings, but you’d be throwing away information.

Focusing on 3 member families is fine—e.g., it is the modal size family, and it usefully eliminates one element of variation in the data. But this does raise that multi-level issue, then—since in groups larger than 2, there may be variation in how structurally embedded individuals are within the group. This issue is partially addressed in Supplementary Tables 21-22 (but, ideally would be done within a multi-level model framework); but the conceptual issue is not.

The analysis R3 proposes does not answer our research question and also has major flaws in our empirical context. The suggested multi-level analysis intends to show how the effects of (triadic) family embeddedness are subject to the variations of individuals’ dyadic embeddedness.

The corresponding model essentially tests the interaction effect of dyadic-triadic embeddedness under different earthquake scenarios. This model setup significantly weakens the identification of our proposed effects that were originally achieved by leveraging an exogenous shock of the earthquake; R3's proposed setting would essentially be a triple interaction between the earthquake and the interaction between dyadic and triadic embeddedness. This triple interaction would make it difficult to identify the impact of embeddedness effects at the triadic structural level (which is our research question).

Moreover, we have already controlled for the effects of dyadic embeddedness in various ways, such as including dyadic embeddedness between the ego and important tie, and family random/fixed effects in various models. R3's proposed analysis will not make our contributions clearer but instead dilute and weaken them, and diminish the paper's readability and clarity.

Of course, some form of multi-level analysis investigating the interaction between dyadic and triadic embeddedness effects could potentially be an extension of our work, but it not germane to our basic research questions.

I guess the biggest question here: what do we think is going on within families that we would expect structural embeddedness to matter? The standard argument in the econ soc literature is that structural embeddedness matters in markets because of concern regarding reputation among third parties. If we have many friends in common, I'm less likely to take advantage of you, because of the reputational ripple effects. But the authors really do not offer much in the way of a theory as to why structural embeddedness would matter for this particular case, other than a general argument that embeddedness matters, and then for intra-family communication, referencing Bott's classic study (which is great, but does not offer insight into why we'd expect embeddedness to matter here).

Our previous manuscript discussed this in the 2nd paragraph of the discussion. We argued that the sharing of social resources matters within families. For example, a family of three where the parents are not friends with their child's friends is different from a family of three where the parents are friends with their child's friends. Clearly the latter structure likely reflects greater information sharing, mutually aligned social interests, greater trust and emotional closeness, etc.

We have added another paragraph to the discussion to clarify our exposition. However, we prefer not to make stronger claims on the 'why' question. Besides being constrained by the article word limit, we are wary of making unverifiable and unfalsifiable claims of social process or mechanism (e.g., we cannot know precisely 'what is going on within families'). While we agree that process questions are always interesting, this is more of a question we wish to pose for future research.

The authors write:

We used the quasi-experimental impact of the earthquake to test whether embeddedness structure and other social network constructs can affect social network dynamics in the aftermath of disaster.

Small point, but better to not write "quasi-experimental impact"—that would be like writing "RCT impact", it just doesn't make sense. It is sensible to say that the earthquake is exogenous, and the severity of the earthquake locally is unrelated to the networks of affected individuals, and that the paper takes advantage of that exogenous variation. A bigger point: it would be recommended for the paper to briefly discuss research design; what type of

quasi-experimental design is it? In the first iteration of the paper, it asserted that it was using a diff-in-diff design, which it was not. Generally the paper is using the earthquake in two ways. The first is that the earthquake is a dramatic change to the context, and allows evaluation of how prior social structure relates to communication in a moment of collective crisis. The second is that the severity of the earthquake varies exogenously, and that that exogenous variation provides local variation in context. So, good to briefly unpack that.

We agree that our study leverages an exogenous and naturally-occurring shock, which we previously may have downplayed. To improve clarity, we now refer to the study instrument (i.e., earthquake) as an exogenous shock throughout the paper. We also agree it is helpful to explicitly emphasize that Models 1-2 utilize earthquake intensity as the interaction term, while Models 3-4 leverage a panel data model (i.e., use, in R3's words, "a dramatic change in context") to study communications behavior changes. We have incorporated these points in the writing. We also comment on a two other points R3 discusses:

- Our original manuscript referred to the analysis of Models 3-4 as a diff-in-diff style analysis (from a statistical model perspective). We already clarified in the last sentence of Methods that:

"Although this analysis is not strictly a difference-in-difference analysis (since we cannot manipulate embeddedness structure), the model specifications are comparable; also, comparisons of the outcome variables between the proposed experiment (fully embedded triad) and control groups (completely unembedded triad) benefit from the exogeneity of the earthquake."

- We do not believe it is necessary to name the statistical paradigm, particularly since we do not make a major contribution to econometrics or statistical methods. Our study is clearly a 'natural experiment', but we eschew such terms out of conservatism; firstly, this avoid over-claiming causality (we recognize there is rarely a perfect statistical instrument); secondly, there is backlash over the term in some social science fields.

Re family members, roaming, etc; I guess the question here is: if these families are cohabiting, and the earthquake hit them when they were home, what were they calling each other about? I.e., I assume that one would expect very different behavioral patterns if this were in the middle of a workday, when people were physically separated. This is a very specific microcontext; it's not clear that any of the referenced literature applies. It is possible, for example, that some people left their homes to go get help, and then had to call their family. Why this might be more likely for an embedded 3 person family than nonembedded is a puzzle.

Roaming means that the user was not in the prefecture (which covers 15,213 km²) at the time of the earthquake. We have noted in the paper, and repeatedly in the 2 previous revision letters, that our effects remain robust when we select only for families that have at least one roaming member. This robustness check helps rule out any physical presence -based explanations (such as cohabitation, someone wandering out to get help, etc.). We have stressed this point repeatedly in the manuscript and previous revision letters.

The citations (which are from survey-based research linking online/phone-based behavior to offline behavior) serve to provide evidence that more non-face-to-face communications is typically correlated with more face-to-face interactions (Eagle et al. 2009; Boase et al., 2006;

Mok et al., 2007) as well as emotional closeness (e.g., Roberts et al. 2009, 2010, Saramäkia et al. 2014).

We cannot investigate offline behaviors or observe (“micro-“) behaviors such as social-coordination (which would require knowing communications content), although we agree this would be a fascinating topic for future study. Such a study would require an extraordinary dataset (and perhaps only possible using survey data).

As for the puzzle, we believe it is more an empirical question than logical puzzle: In this paper, we show that fully embedded families are more likely to 1) communicate with each other, 2) reciprocate each other’s calls, and 3) mobilize their joint social networks. Based on these findings, we would also expect that fully embedded families should share relatively more social resources in general (whether they are online or face-to-face): One might expect this to also correspond with greater cooperation and coordination behaviors. Again, this is a question for future research.

We conclude by thanking R3 for their comments; although we disagree with many of them, we believe we have strengthened the paper and our research in responding to them.

References

1. Benson, A., Abebe, R., Schaub, M., Jadbabaie, A., Kleinberg, J. Simplicial Closure and Higher-order Link Prediction. *Proc Natl Acad Sci USA* **115**, 11221-11230 (2018)
2. Lambiotte, R., Rosvall, M., Scholtes, I. From networks to optimal higher-order models of complex systems. *Nat Phys* **15**, 313–320 (2019)
3. Milo, R. et al. Network motifs: Simple building blocks of complex networks. *Science* **298**, 824–827 (2002).
4. Milo, R. et al. Super families of evolved and designed networks. *Science* **303**, 1538–1542 (2004).
5. Kovanen, L., Kaski, K., Kertész, J., Saramäki, J. Temporal motifs reveal homophily, gender-specific patterns, and group talk in call sequences. *Proc Natl Acad Sci USA* **110**, 18070-18075 (2013).
6. Roberts, S. B. G. & Dunbar, R. I. M. Communication in social networks: effects of kinship, network size and emotional closeness. *Pers. Relationships* **18**, 439–452 (2010).
7. Roberts, S. B. G., Dunbar, R. I. M., Pollet, T. & Kuppens, T. Exploring variations in active network size: constraints and ego characteristics. *Social Networks* **31**, 138–146 (2009).
8. Saramäkia, J., Leichtb, E. A., Lópezb, E., Roberts, S. G. B., Reed-Tsochasb, F., Dunbar, R. I. M. Persistence of social signatures in human communication. *Proc Natl Acad Sci USA* **111**, 942–947 (2014).
9. Eagle, N., Pentland, A., Lazer, D. Inferring friendship network structure using mobile phone data. *Proc Natl Acad Sci USA* **106**, 15274-15278 (2009).
10. Boase, J., Horrigan, J.B., Wellman, B., Rainie, L. *The strength of Internet ties. Pew Internet & American Life Project* (2006). Retrieved from: <https://www.umass.edu/digitalcenter/sites/default/files/PIP%20Internet%20ties.pdf>
11. Mok, D., Wellman, B., Basu, R. Did distance matter before the Internet? Interpersonal contact and support in the 1970s. *Social Networks* **29**, 430–461 (2007).

REVIEWERS' COMMENTS

Reviewer #4 (Remarks to the Author):

There is clearly nothing wrong with this MS. Why are you wasting busy academics' time with (a) endless review cycles and (b) endless reviewing. This should be done in-house. Make your own mind up and dont be so damn lazy.